# Sampling is as easy as learning the score: theory for diffusion models with minimal data assumptions

**Sitan Chen**[*] **Sinho Chewi**[†] **Jerry Li**[‡] **Yuanzhi Li**[§] **Adil Salim**[¶] **Anru R. Zhang**[‖]

## Abstract

We provide theoretical convergence guarantees for score-based generative models (SGMs) such as denoising diffusion probabilistic models (DDPMs), which constitute the backbone of large-scale real-world generative models such as DALL·E 2. Our main result is that, assuming accurate score estimates, such SGMs can efficiently sample from essentially any realistic data distribution. In contrast to prior works, our results (1) hold for an $L^2$-accurate score estimate (rather than $L^\infty$-accurate); (2) do not require restrictive functional inequality conditions that preclude substantial non-log-concavity; (3) scale polynomially in all relevant problem parameters; and (4) match state-of-the-art complexity guarantees for discretization of the Langevin diffusion, provided that the score error is sufficiently small. We view this as strong theoretical justification for the empirical success of SGMs. We also examine SGMs based on the critically damped Langevin diffusion (CLD). Contrary to conventional wisdom, we provide evidence that the use of the CLD does *not* reduce the complexity of SGMs.

## 1 Introduction

Score-based generative models (SGMs) are a family of generative models which achieve state-of-the-art performance for generating audio and image data (Sohl-Dickstein et al., 2015; Ho et al., 2020; Dhariwal & Nichol, 2021; Kingma et al., 2021; Song et al., 2021a;b; Vahdat et al., 2021); see, e.g., the recent surveys (Cao et al., 2022; Croitoru et al., 2022; Yang et al., 2022). For example, denoising diffusion probabilistic models (DDPMs) (Sohl-Dickstein et al., 2015; Ho et al., 2020) are a key component in large-scale generative models such as DALL·E 2 (Ramesh et al., 2022). As the importance of SGMs continues to grow due to newfound applications in commercial domains, it is a pressing question of both practical and theoretical concern to understand the mathematical underpinnings which explain their startling empirical successes.

As we explain in Section 2, at their mathematical core, SGMs consist of two stochastic processes, the forward process and the reverse process. The forward process transforms samples from a data distribution $q$ (e.g., images) into noise, whereas the reverse process transforms noise into samples from $q$, hence performing generative modeling. Running the reverse process requires estimating the *score function* of the law of the forward process; this is typically done by training neural networks on a score matching objective (Hyvärinen, 2005; Vincent, 2011; Song & Ermon, 2019).

Providing precise guarantees for estimation of the score function is difficult, as it requires an understanding of the non-convex training dynamics of neural network optimization that is currently out of reach. However, given the empirical success of neural networks on the score estimation task,

---

[*]Department of EECS at University of California, Berkeley, `sitan@seas.harvard.edu`.

[†]Department of Mathematics at Massachusetts Institute of Technology, `schewi@mit.edu`. Part of this work was done while SC was a research intern at Microsoft Research.

[‡]Microsoft Research, `jerrl@microsoft.com`.

[§]Microsoft Research and Machine Learning Department at Carnegie Mellon University, `yuanzhil@andrew.cmu.edu`.

[¶]Microsoft Research, `adilsalim@microsoft.com`.

[‖]Departments of Biostatistics & Bioinformatics, Computer Science, Mathematics, and Statistical Science at Duke University, `anru.zhang@duke.edu`.

a natural and important question is whether accurate score estimation implies that SGMs provably converge to the true data distribution in realistic settings. This is a surprisingly delicate question, as even with accurate score estimates, as we explain in Section 2.1, there are several other sources of error which could cause the SGM to fail to converge. Indeed, despite a flurry of recent work (Block et al., 2020; De Bortoli et al., 2021; De Bortoli, 2022; Lee et al., 2022a; Pidstrigach, 2022; Liu et al., 2022), prior analyses fall short of answering this question, for (at least) one of three main reasons:

1. **Super-polynomial convergence.** The bounds obtained are not quantitative (e.g., De Bortoli et al., 2021; Pidstrigach, 2022), or scale exponentially in the dimension and other problem parameters like time and smoothness (Block et al., 2020; De Bortoli, 2022; Liu et al., 2022), and hence are typically vacuous for the high-dimensional settings of interest in practice.
2. **Strong assumptions on the data distribution.** The bounds require strong assumptions on the true data distribution, such as a log-Sobelev inequality (LSI) (see, e.g., Lee et al., 2022a). While the LSI is slightly weaker than strong log-concavity, it ultimately precludes the presence of substantial non-convexity, which impedes the application of these results to complex and highly multi-modal real-world data distributions. Indeed, obtaining a polynomial-time convergence analysis for SGMs that holds for multi-modal distributions was posed as an open question in (Lee et al., 2022a).
3. **Strong assumptions on the score estimation error.** The bounds require that the score estimate is $L^\infty$-accurate (i.e., *uniformly* accurate), as opposed to $L^2$-accurate (see, e.g., De Bortoli et al., 2021). This is problematic because the score matching objective is an $L^2$ loss (see Section A.1 in the supplement), and there are empirical studies suggesting that in practice, the score estimate is not in fact $L^\infty$-accurate (e.g., Zhang & Chen, 2022). Intuitively, this is because we cannot expect that the score estimate we obtain will be accurate in regions of space where the true density is very low, simply because we do not expect to see many (or indeed, any) samples from there.

Providing an analysis which goes beyond these limitations is a pressing first step towards theoretically understanding why SGMs actually work in practice.

## 1.1 OUR CONTRIBUTIONS

In this work, we take a step towards bridging theory and practice by providing a convergence guarantee for SGMs, under realistic (in fact, quite minimal) assumptions, which scales polynomially in all relevant problem parameters. Namely, our main result (Theorem 2) only requires the following assumptions on the data distribution $q$, which we make more quantitative in Section 3:

**A1** The score function of the forward process is $L$-Lipschitz.

**A2** The $(2 + \eta)$-th moment of $q$ is finite, where $\eta > 0$ is an arbitrarily small constant.

**A3** The data distribution $q$ has finite KL divergence w.r.t. the standard Gaussian.

We note that all of these assumptions are either standard or, in the case of **A2**, far weaker than what is needed in prior work. Crucially, unlike prior works, we do *not* assume log-concavity, an LSI, or dissipativity; hence, our assumptions cover *arbitrarily non-log-concave* data distributions. Our main result is summarized informally as follows.

**Theorem 1** (informal, see Theorem 2). *Under assumptions* **A1-A3**, *and if the score estimation error in $L^2$ is at most $\widetilde{O}(\varepsilon)$, then with an appropriate choice of step size, the SGM outputs a measure which is $\varepsilon$-close in total variation (TV) distance to $q$ in $\widetilde{O}(L^2 d/\varepsilon^2)$ iterations.*

Our iteration complexity is quite tight: it matches state-of-the-art discretization guarantees for the Langevin diffusion (Vempala & Wibisono, 2019; Chewi et al., 2021a).

We find Theorem 1 surprising, because it shows that SGMs can sample from the data distribution $q$ with polynomial complexity, even when $q$ is highly non-log-concave (a task that is usually intractable), *provided that one has access to an accurate score estimator*. This answers the open question of (Lee et al., 2022a) regarding whether or not SGMs can sample from multimodal distributions, e.g., mixtures of distributions with bounded log-Sobolev constant. In the context of neural networks, our result implies that so long as the neural network succeeds at the score estimation task, the remaining part of the SGM algorithm based on the diffusion model is completely principled, in that it admits a strong theoretical justification.

In general, learning the score function is also a difficult task. Nevertheless, our result opens the door to further investigations, such as: do score functions for real-life data have intrinsic (e.g., low-dimensional) structure which can be exploited by neural networks? A positive answer to this question, combined with our sampling result, would then provide an end-to-end guarantee for SGMs.

More generally, our result can be viewed as a black-box reduction of the task of sampling to the task of learning the score function of the forward process, at least for distributions satisfying our mild assumptions. Existing computational hardness results for learning natural high-dimensional distributions like mixtures of Gaussians (Diakonikolas et al., 2017; Bruna et al., 2021; Gupte et al., 2022) and pushforwards of Gaussians by shallow ReLU networks (Daniely & Vardi, 2021; Chen et al., 2022a;b) thus immediately imply hardness of score estimation for these distributions. To our knowledge this yields the first known information-computation gaps for this task.

**Arbitrary distributions with bounded support.** The assumption that the score function is Lipschitz entails in particular that the data distribution has a density w.r.t. Lebesgue measure; in particular, our theorem fails when $q$ satisfies the manifold hypothesis, i.e., is supported on a lower-dimensional submanifold of $\mathbb{R}^d$. But this is for good reason: it is not possible to obtain non-trivial TV guarantees, because the output distribution of the SGM has full support. Instead, we show in Section 3.2 that we can obtain polynomial convergence guarantees in the bounded Lipschitz metric by stopping the SGM algorithm early, or in the Wasserstein metric by an additional truncation step, under the *sole* assumption that the data distribution $q$ has bounded support, without assuming that $q$ has a density. Since data distributions encountered in real life satisfy this assumption, our results yield the following compelling takeaway:

*Given an $L^2$-accurate score estimate, SGMs can sample from (essentially) any data distribution*.

**Critically damped Langevin diffusion (CLD).** Using our techniques, we also investigate the use of the critically damped Langevin diffusion (CLD) for SGMs, which was proposed in (Dockhorn et al., 2022). Although numerical experiments and intuition from the log-concave sampling literature suggest that the CLD could potentially speed up sampling via SGMs, we provide theoretical evidence to the contrary. Based on this, in Section 3.3, we conjecture that SGMs based on the CLD do not exhibit improved dimension dependence compared to the original DDPM algorithm.

## 1.2 PRIOR WORK

We now provide a detailed comparison to prior work. By now, there is a vast literature on providing precise complexity estimates for log-concave sampling; see, e.g., Chewi (2022) for an exposition on recent developments. The proofs in this work build upon the techniques developed in this literature. However, our work addresses the significantly more challenging setting of *non-log-concave* sampling.

The work of De Bortoli et al. (2021) provides guarantees for the diffusion Schrödinger bridge (Song et al., 2021b). However, as previously mentioned their result is not quantitative, and they require an $L^\infty$-accurate score estimate. The works Block et al. (2020); Lee et al. (2022a); Liu et al. (2022) instead analyze SGMs under the more realistic assumption of an $L^2$-accurate score estimate. However, the bounds of Block et al. (2020); Liu et al. (2022) suffer from exponential dependencies on parameters like dimension and smoothness, whereas the bounds of Lee et al. (2022a) require $q$ to satisfy an LSI.

The recent work of De Bortoli (2022), motivated by the *manifold hypothesis*, considers a different pointwise assumption on the score estimation error which allows the error to blow up at time $0$ and at spatial $\infty$. We discuss the manifold setting in more detail in Section 3.2. Unfortunately, the bounds of De Bortoli (2022) also scale exponentially in problem parameters such as the manifold diameter.

We also mention that the use of reversed SDEs for sampling is implicit in the interpretation of the proximal sampler (Lee et al., 2021) given by Chen et al. (2022c). Our work can be viewed as expanding upon the theory of Chen et al. (2022c) using a different forward channel (the OU process).

**Concurrent work.** Very recently, Lee et al. (2022b) independently obtained results similar to our results for DDPM. While our assumptions are technically somewhat incomparable (they assume the score error can vary with time but assume the data is compactly supported), our quantitative bounds are stronger. Additionally, the upper and lower bounds for CLD are unique to our work.

## 2 BACKGROUND ON SGMS

Throughout this paper, given a probability measure $p$ which admits a density w.r.t. Lebesgue measure, we abuse notation and identify it with its density function. Additionally, we will let $q$ denote the data distribution from which we want to generate new samples. We assume that $q$ is a probability measure on $\mathbb{R}^d$ with full support, and that it admits a smooth density written $q = \exp(-U)$ (we relax this assumption in Section 3.2).

In this section, we provide a brief exposition to SGMs, following Song et al. (2021b).

### 2.1 BACKGROUND ON DENOISING DIFFUSION PROBABILISTIC MODELING (DDPM)

**Forward process.** In denoising diffusion probabilistic modeling (DDPM), we start with a forward process, which is a stochastic differential equation (SDE). For clarity, we consider the simplest possible choice, which is the Ornstein–Uhlenbeck (OU) process

$$\mathrm{d}\bar{X}_t = -\bar{X}_t \, \mathrm{d}t + \sqrt{2} \, \mathrm{d}B_t \,, \qquad \bar{X}_0 \sim q \,, \tag{1}$$

where $(B_t)_{t \geq 0}$ is a standard Brownian motion in $\mathbb{R}^d$. The OU process is the unique time-homogeneous Markov process which is also a Gaussian process, with stationary distribution equal to the standard Gaussian distribution $\gamma^d$ on $\mathbb{R}^d$. In practice, it is also common to introduce a positive smooth function $g : \mathbb{R}_+ \to \mathbb{R}$ and consider the time-rescaled OU process

$$\mathrm{d}\bar{X}_t = -g(t)^2 \, \bar{X}_t \, \mathrm{d}t + \sqrt{2} \, g(t) \, \mathrm{d}B_t \,, \qquad X_0 \sim q \,. \tag{2}$$

Although our analysis could be extended to consider these variants, in this work we stick with the choice $g \equiv 1$ for simplicity; see Song et al. (2021b) for further discussion.

The forward process has the interpretation of transforming samples from the data distribution $q$ into pure noise. From the well-developed theory of Markov diffusions, it is known that if $q_t := \mathrm{law}(X_t)$ denotes the law of the OU process at time $t$, then $q_t \to \gamma^d$ exponentially fast in various divergences and metrics such as the 2-Wasserstein metric $W_2$; see Bakry et al. (2014).

**Reverse process.** If we reverse the forward process (1) in time, then we obtain a process that transforms noise into samples from $q$, which is the aim of generative modeling. In general, suppose that we have an SDE of the form

$$\mathrm{d}\bar{X}_t = b_t(\bar{X}_t) \, \mathrm{d}t + \sigma_t \, \mathrm{d}B_t \,,$$

where $(\sigma_t)_{t \geq 0}$ is a deterministic matrix-valued process. Then, under mild conditions on the process (e.g., Föllmer, 1985; Cattiaux et al., 2022), which are satisfied for all processes under consideration in this work, the reverse process also admits an SDE description. Namely, if we fix the terminal time $T > 0$ and set

$$\bar{X}_t^{\leftarrow} := \bar{X}_{T-t} \,, \qquad \text{for } t \in [0, T] \,,$$

then the process $(\bar{X}_t^{\leftarrow})_{t \in [0,T]}$ satisfies the SDE

$$\mathrm{d}\bar{X}_t^{\leftarrow} = b_t^{\leftarrow}(\bar{X}_t^{\leftarrow}) \, \mathrm{d}t + \sigma_{T-t} \, \mathrm{d}B_t \,,$$

where the backwards drift satisfies the relation

$$b_t + b_{T-t}^{\leftarrow} = \sigma_t \sigma_t^{\mathsf{T}} \nabla \ln q_t \,, \qquad q_t := \mathrm{law}(\bar{X}_t) \,. \tag{3}$$

Applying this to the forward process (1), we obtain the reverse process

$$\mathrm{d}\bar{X}_t^{\leftarrow} = \{\bar{X}_t^{\leftarrow} + 2 \nabla \ln q_{T-t}(\bar{X}_t^{\leftarrow})\} \, \mathrm{d}t + \sqrt{2} \, \mathrm{d}B_t \,, \qquad \bar{X}_0^{\leftarrow} \sim q_T \,, \tag{4}$$

where now $(B_t)_{t \in [0,T]}$ is the reversed Brownian motion.[1] Here, $\nabla \ln q_t$ is called the *score function* for $q_t$. Since $q$ (and hence $q_t$ for $t \geq 0$) is not explicitly known, in order to implement the reverse process the score function must be estimated on the basis of samples. The mechanism behind this is the idea of *score matching* which goes back to Hyvärinen (2005); Vincent (2011): roughly speaking,

---

[1] For ease of notation, we do not distinguish between the forward and the reverse Brownian motions.

Gaussian integration by parts implies that minimizing the $L^2(q_t)$ loss achieved by an estimate $s_t$ for the score $\nabla \ln q_t$ is *equivalent* to minimizing the $L^2(q_t)$ loss in predicting, given a sample from the forward process at time $t$, what noise was applied to the corresponding sample at time $0$ to obtain it. We defer an exposition of the details of score matching to Sections A.1 and D of the supplement.

In light of this, it is thus most natural to assume an $L^2(q_t)$ error bound $\mathbb{E}_{q_t}[\|s_t - \nabla \ln q_t\|^2] \le \varepsilon_{\text{score}}^2$ for the score estimator $s_t$. If $s_t$ is taken to be the empirical risk minimizer for a suitable function class, then guarantees for the $L^2(q_t)$ error can be obtained via standard statistical analysis, see, e.g., Block et al. (2020).

**Discretization and implementation.** We now discuss the final steps required to obtain an implementable algorithm. First, in the learning phase, given samples $\bar{X}_0^{(1)}, \dots, \bar{X}_0^{(n)}$ from $q$ (e.g., a database of natural images), we train a neural network via score matching, see Song & Ermon (2019). Let $h > 0$ be the step size of the discretization; we assume that we have obtained a score estimate $s_{kh}$ of $\nabla \ln q_{kh}$ for each time $k = 0, 1, \dots, N$, where $T = Nh$.

In order to approximately implement the reverse SDE (4), we first replace the score function $\nabla \ln q_{T-t}$ with the estimate $s_{T-t}$. Then, for $t \in [kh, (k+1)h]$ we freeze the value of this coefficient in the SDE at time $kh$. It yields the new SDE

$$\mathrm{d}X_t^{\leftarrow} = \{X_t^{\leftarrow} + 2\, s_{T-kh}(X_{kh}^{\leftarrow})\}\, \mathrm{d}t + \sqrt{2}\, \mathrm{d}B_t\,, \qquad t \in [kh, (k+1)h]\,. \tag{5}$$

Since this is a linear SDE, it can be integrated in closed form; in particular, conditionally on $X_{kh}^{\leftarrow}$, the next iterate $X_{(k+1)h}^{\leftarrow}$ has an explicit Gaussian distribution.

There is one final detail: although the reverse SDE (4) should be started at $q_T$, we do not have access to $q_T$ directly. Instead, taking advantage of the fact that $q_T \approx \gamma^d$, we instead initialize the algorithm at $X_0^{\leftarrow} \sim \gamma^d$, i.e., from pure noise.

Let $p_t := \mathrm{law}(X_t^{\leftarrow})$ denote the law of the algorithm at time $t$. The goal of this work is to bound $\mathsf{TV}(p_T, q)$, taking into account three sources of error: (1) estimation of the score; (2) discretization of the SDE with step size $h > 0$; and (3) initialization of the algorithm at $\gamma^d$ rather than at $q_T$.

## 2.2 Background on the critically damped Langevin diffusion (CLD)

The critically damped Langevin diffusion (CLD) is based on the forward process

$$\begin{aligned} \mathrm{d}\bar{X}_t &= -\bar{V}_t\, \mathrm{d}t\,, \\ \mathrm{d}\bar{V}_t &= -(\bar{X}_t + 2\,\bar{V}_t)\, \mathrm{d}t + 2\, \mathrm{d}B_t\,. \end{aligned} \tag{6}$$

Compared to the OU process (1), this is now a coupled system of SDEs, where we have introduced a new variable $\bar{V}$ representing the velocity process. The stationary distribution of the process is $\gamma^{2d}$, the standard Gaussian measure on phase space $\mathbb{R}^d \times \mathbb{R}^d$, and we initialize at $\bar{X}_0 \sim q$ and $\bar{V}_0 \sim \gamma^d$.

More generally, the CLD (6) is an instance of what is referred to as the *kinetic Langevin* or the *underdamped Langevin* process in the sampling literature. In the context of strongly log-concave sampling, the smoother paths of $\bar{X}$ lead to smaller discretization error, thereby furnishing an algorithm with $\widetilde{O}(\sqrt{d}/\varepsilon)$ gradient complexity (as opposed to sampling based on the overdamped Langevin process, which has complexity $\widetilde{O}(d/\varepsilon^2)$), see Cheng et al. (2018); Shen & Lee (2019); Dalalyan & Riou-Durand (2020); Ma et al. (2021). The recent paper Dockhorn et al. (2022) proposed to use the CLD as the basis for an SGM and they empirically observed improvements over DDPM.

Applying (3), the corresponding reverse process is

$$\begin{aligned} \mathrm{d}\bar{X}_t^{\leftarrow} &= -\bar{V}_t^{\leftarrow}\, \mathrm{d}t\,, \\ \mathrm{d}\bar{V}_t^{\leftarrow} &= \big(\bar{X}_t^{\leftarrow} + 2\,\bar{V}_t^{\leftarrow} + 4\, \nabla_v \ln \boldsymbol{q}_{T-t}(\bar{X}_t^{\leftarrow}, \bar{V}_t^{\leftarrow})\big)\, \mathrm{d}t + 2\, \mathrm{d}B_t\,, \end{aligned} \tag{7}$$

where $\boldsymbol{q}_t := \mathrm{law}(\bar{X}_t, \bar{V}_t)$ is the law of the forward process at time $t$. Note that the gradient in the score function is only taken w.r.t. the velocity coordinate. Upon replacing the score function with an estimate $\boldsymbol{s}$, we arrive at the algorithm

$$\begin{aligned} \mathrm{d}X_t^{\leftarrow} &= -V_t^{\leftarrow}\, \mathrm{d}t\,, \\ \mathrm{d}V_t^{\leftarrow} &= \big(X_t^{\leftarrow} + 2\,V_t^{\leftarrow} + 4\, \boldsymbol{s}_{T-kh}(X_{kh}^{\leftarrow}, V_{kh}^{\leftarrow})\big)\, \mathrm{d}t + 2\, \mathrm{d}B_t\,, \end{aligned}$$

for $t \in [kh, (k+1)h]$. We provide further background on the CLD in Section C.1.

## 3 RESULTS

We now formally state our assumptions and our main results.

### 3.1 RESULTS FOR DDPM

For DDPM, we make the following mild assumptions on the data distribution $q$.

**Assumption 1** (Lipschitz score). *For all $t \geq 0$, the score $\nabla \ln q_t$ is $L$-Lipschitz.*

**Assumption 2** (second moment bound). *For some $\eta > 0$, $\mathbb{E}_q[\|\cdot\|^{2+\eta}]$ is finite. We also write $\mathfrak{m}_2^2 := \mathbb{E}_q[\|\cdot\|^2]$ for the second moment of $q$.*

For technical reasons, we need to assume that $q$ has a finite moment of order slightly but strictly bigger than 2, but our quantitative bounds will only depend on the second moment $\mathfrak{m}_2^2$.

Assumption 1 is standard and has been used in the prior works Block et al. (2020); Lee et al. (2022a). However, unlike Lee et al. (2022a); Liu et al. (2022), we do not assume Lipschitzness of the score estimate. Moreover, unlike Block et al. (2020); De Bortoli et al. (2021); Liu et al. (2022), we do not assume any convexity or dissipativity assumptions on the potential $U$, and unlike Lee et al. (2022a) we do not assume that $q$ satisfies a log-Sobolev inequality. Hence, our assumptions cover a wide range of highly non-log-concave data distributions. Our proof technique is fairly robust and even Assumption 1 could be relaxed (as well as other extensions, such as considering the time-changed forward process (2)), although we focus on the simplest setting in order to better illustrate the conceptual significance of our results.

We also assume a bound on the score estimation error.

**Assumption 3** (score estimation error). *For all $k = 1, \ldots, N$, $\mathbb{E}_{q_{kh}}[\|s_{kh} - \nabla \ln q_{kh}\|^2] \leq \varepsilon_{\mathrm{score}}^2$.*

This is the same assumption as in Lee et al. (2022a), and as discussed in Section 2.1, it is a natural and realistic assumption in light of the derivation of the score matching objective.

Our main result for DDPM is the following theorem.

**Theorem 2** (DDPM, see Section B of supplement). *Suppose that Assumptions 1, 2, and 3 hold. Let $p_T$ be the output of the DDPM algorithm (Section 2.1) at time $T$, and suppose that the step size $h := T/N$ satisfies $h \lesssim 1/L$, where $L \geq 1$. Then, it holds that*

$$\mathsf{TV}(p_T, q) \lesssim \underbrace{\sqrt{\mathsf{KL}(q \,\|\, \gamma^d)} \exp(-T)}_{\textit{convergence of forward process}} + \underbrace{(L\sqrt{dh} + L\mathfrak{m}_2 h)\sqrt{T}}_{\textit{discretization error}} + \underbrace{\varepsilon_{\mathrm{score}}\sqrt{T}}_{\textit{score estimation error}} .$$

To interpret this result, suppose that, e.g., $\mathsf{KL}(q \,\|\, \gamma^d) \leq \mathrm{poly}(d)$ and $\mathfrak{m}_2 \leq d$.[2] Choosing $T \asymp \log(\mathsf{KL}(q \,\|\, \gamma)/\varepsilon)$ and $h \asymp \frac{\varepsilon^2}{L^2 d}$, and hiding logarithmic factors,

$$\mathsf{TV}(p_T, q) \leq \widetilde{O}(\varepsilon + \varepsilon_{\mathrm{score}}), \qquad \text{for } N = \widetilde{\Theta}\Big(\frac{L^2 d}{\varepsilon^2}\Big).$$

In particular, in order to have $\mathsf{TV}(p_T, q) \leq \varepsilon$, it suffices to have score error $\varepsilon_{\mathrm{score}} \leq \widetilde{O}(\varepsilon)$.

We remark that the iteration complexity of $N = \widetilde{\Theta}(\frac{L^2 d}{\varepsilon^2})$ matches state-of-the-art complexity bounds for the Langevin Monte Carlo (LMC) algorithm for sampling under a log-Sobolev inequality (LSI), see Vempala & Wibisono (2019); Chewi et al. (2021a). This provides some evidence that our discretization bounds are of the correct order, at least with respect to the dimension and accuracy parameters, and without higher-order smoothness assumptions.

### 3.2 CONSEQUENCES FOR ARBITRARY DATA DISTRIBUTIONS WITH BOUNDED SUPPORT

We now elaborate upon the implications of our results under the *sole* assumption that the data distribution $q$ is compactly supported, $\mathrm{supp}\, q \subseteq \mathsf{B}(0, R)$. In particular, we do not assume that $q$ has a

---

[2]For many distributions of interest, e.g., the standard Gaussian distribution or product measures, in fact we have $\mathfrak{m}_2 = O(\sqrt{d})$. Also, for applications to images in which each coordinate (i.e., pixel) lies in a bounded range $[-1, 1]$, we also have $\mathfrak{m}_2 \leq \sqrt{d}$.

smooth density w.r.t. Lebesgue measure, which allows for studying the case when $q$ is supported on a lower-dimensional submanifold of $\mathbb{R}^d$ as in the *manifold hypothesis*. This setting was investigated recently in De Bortoli (2022).

For this setting, our results do not apply directly because the score function of $q$ is not well-defined and hence Assumption 1 fails to hold. Also, the bound in Theorem 2 has a term involving $\mathsf{KL}(q \parallel \gamma^d)$ which is infinite if $q$ is not absolutely continuous w.r.t. $\gamma^d$. As pointed out by De Bortoli (2022), in general we cannot obtain non-trivial guarantees for $\mathsf{TV}(p_T, q)$, because $p_T$ has full support and therefore $\mathsf{TV}(p_T, q) = 1$ under the manifold hypothesis. Nevertheless, we show that we can apply our results using an early stopping technique.

Namely, using the following lemma, we obtain a sequence of corollaries.

**Lemma 3** (Lemma 21 in supplement). *Suppose that* $\mathrm{supp}\, q \subseteq \mathsf{B}(0, R)$ *where* $R \geq 1$, *and let* $q_t$ *denote the law of the OU process at time* $t$, *started at* $q$. *Let* $\varepsilon_{W_2} > 0$ *be such that* $\varepsilon_{W_2} \ll \sqrt{d}$ *and set* $t \asymp \varepsilon_{W_2}^2/(\sqrt{d}\,(R \vee \sqrt{d}))$. *Then, (1)* $W_2(q_t, q) \leq \varepsilon_{W_2}$, *(2)* $q_t$ *satisfies* $\mathsf{KL}(q_t \parallel \gamma^d) \lesssim \frac{\sqrt{d}\,(R \vee \sqrt{d})^3}{\varepsilon_{W_2}^2}$, *and (3) for every* $t' \geq t$, $q_{t'}$ *satisfies Assumption 1 with* $L \lesssim \frac{dR^2\,(R \vee \sqrt{d})^2}{\varepsilon_{W_2}^4}$.

By substituting $q_t$ for this choice of $t$ in place of $q$ in Theorem 2, we obtain Corollary 4 below. We remark that taking $q_t$ as the new target corresponds to stopping the algorithm early: instead of running the algorithm backward for a time $T$, we run the algorithm backward for a time $T - t$ (note that $T - t$ should be a multiple of the step size $h$).

**Corollary 4** (compactly supported data). *Suppose that* $q$ *is supported on the ball of radius* $R \geq 1$. *Let* $t \asymp \varepsilon_{W_2}^2/(\sqrt{d}\,(R \vee \sqrt{d}))$. *Then, the output* $p_{T-t}$ *of DDPM is* $\varepsilon_{\mathrm{TV}}$-*close in TV to the distribution* $q_t$, *which is* $\varepsilon_{W_2}$-*close in* $W_2$ *to* $q$, *provided that the step size* $h$ *is chosen appropriately according to Theorem 2 and* $N = \widetilde{\Theta}\Big(\frac{d^3 R^4\,(R \vee \sqrt{d})^4}{\varepsilon_{\mathrm{TV}}^2\,\varepsilon_{W_2}^8}\Big)$ *and* $\varepsilon_{\mathrm{score}} \leq \widetilde{O}(\varepsilon_{\mathrm{TV}})$.

Observing that both the TV and $W_1$ metrics are upper bounds for the bounded Lipschitz metric $\mathsf{d}_{\mathrm{BL}}(\mu, \nu) := \sup\{\int f \,\mathrm{d}\mu - \int f \,\mathrm{d}\nu \mid f : \mathbb{R}^d \to [-1, 1] \text{ is 1-Lipschitz}\}$, we immediately obtain the following corollary.

**Corollary 5** (compactly supported data, BL metric). *Suppose that* $q$ *is supported on the ball of radius* $R \geq 1$. *Let* $t \asymp \varepsilon_{W_2}^2/(\sqrt{d}\,(R \vee \sqrt{d}))$. *Then, the output* $p_{T-t}$ *of the DDPM algorithm satisfies* $\mathsf{d}_{\mathrm{BL}}(p_{T-t}, q) \leq \varepsilon$, *provided that the step size* $h$ *is chosen appropriately according to Theorem 2 and* $N = \widetilde{\Theta}(d^3 R^4\,(R \vee \sqrt{d})^4/\varepsilon^{10})$ *and* $\varepsilon_{\mathrm{score}} \leq \widetilde{O}(\varepsilon_{\mathrm{TV}})$.

Finally, if the output $p_{T-t}$ of DDPM at time $T - t$ is projected onto $\mathsf{B}(0, R_0)$ for an appropriate choice of $R_0$, then we can also translate our guarantees to the standard $W_2$ metric, which we state as the following corollary.

**Corollary 6** (compactly supported data, $W_2$ metric; see Section B.5 in supplement). *Suppose that* $q$ *is supported on the ball of radius* $R \geq 1$. *Let* $t \asymp \varepsilon_{W_2}^2/(\sqrt{d}\,(R \vee \sqrt{d}))$, *and let* $p_{T-t, R_0}$ *denote the output of DDPM at time* $T - t$ *projected onto* $\mathsf{B}(0, R_0)$ *for* $R_0 = \widetilde{\Theta}(R)$. *Then, it holds that* $W_2(p_{T-t, R_0}, q) \leq \varepsilon$, *provided that the step size* $h$ *is chosen appropriately according to Theorem 2,* $N = \widetilde{\Theta}(d^3 R^8\,(R \vee \sqrt{d})^4/\varepsilon^{12})$, *and* $\varepsilon_{\mathrm{score}} \leq \widetilde{O}(\varepsilon_{\mathrm{TV}})$.

Note that the dependencies in the three corollaries above are polynomial in all of the relevant problem parameters. In particular, since the last corollary holds in the $W_2$ metric, it is directly comparable to De Bortoli (2022) and vastly improves upon the exponential dependencies therein.

### 3.3 Results for CLD

In order to state our results for score-based generative modeling based on the CLD, we must first modify Assumptions 1 and 3 accordingly.

**Assumption 4.** *For all* $t \geq 0$, *the score* $\nabla_v \ln \boldsymbol{q}_t$ *is* $L$-*Lipschitz.*

**Assumption 5.** *For all* $k = 1, \ldots, N$, $\mathbb{E}_{\boldsymbol{q}_{kh}}[\|\boldsymbol{s}_{kh} - \nabla_v \ln \boldsymbol{q}_{kh}\|^2] \leq \varepsilon_{\mathrm{score}}^2$.

If we ignore the dependence on $L$ and assume that the score estimate is sufficiently accurate, then the iteration complexity guarantee of Theorem 2 is $N = \widetilde{\Theta}(d/\varepsilon^2)$. On the other hand, recall from Section 2.2 that based on intuition from the literature on log-concave sampling and from empirical findings in Dockhorn et al. (2022), we might expect that SGMs based on the CLD have a smaller iteration complexity than DDPM. We prove the following theorem.

**Theorem 7** (CLD, see Section C of supplement). *Suppose that Assumptions 2, 4, and 5 hold. Let $\boldsymbol{p}_T$ be the output of the SGM algorithm based on the CLD (Section 2.2) at time $T$, and suppose the step size $h := T/N$ satisfies $h \lesssim 1/L$, where $L \geq 1$. Then, there is a universal constant $c > 0$ such that $\mathsf{TV}(\boldsymbol{p}_T, q \otimes \gamma^d)$ is bounded, up to a constant factor, by*

$$\underbrace{\sqrt{\mathsf{KL}(q \parallel \gamma^d) + \mathsf{FI}(q \parallel \gamma^d)} \exp(-cT)}_{\text{convergence of forward process}} + \underbrace{(L\sqrt{dh} + L\mathfrak{m}_2 h)\sqrt{T}}_{\text{discretization error}} + \underbrace{\varepsilon_{\text{score}}\sqrt{T}}_{\text{score estimation error}} \, ,$$

*where $\mathsf{FI}(q \parallel \gamma^d)$ is the relative Fisher information $\mathsf{FI}(q \parallel \gamma^d) := \mathbb{E}_q[\|\nabla \ln(q/\gamma^d)\|^2]$.*

Note that the result of Theorem 7 is in fact no better than our guarantee for DDPM in Theorem 2. Although it is possible that this is an artefact of our analysis, we believe that it is in fact fundamental. As we discuss in Remark C.2, from the form of the reverse process (7), the SGM based on CLD lacks a certain property (that the discretization error should only depend on the size of the increment of the $X$ process, not the increments of both the $X$ and $V$ processes) which is crucial for the improved dimension dependence of the CLD over the Langevin diffusion in log-concave sampling. Hence, in general, we conjecture that under our assumptions, SGMs based on the CLD do not achieve a better dimension dependence than DDPM.

We provide evidence for our conjecture via a lower bound. In our proofs of Theorems 2 and 7, we rely on bounding the KL divergence between certain measures on the path space $\mathcal{C}([0, T]; \mathbb{R}^d)$ via Girsanov's theorem. The following result lower bounds this KL divergence, even for the setting in which the score estimate is perfect ($\varepsilon_{\text{score}} = 0$) and the data distribution $q$ is the standard Gaussian.

**Theorem 8** (Section C.5 of supplement). *Let $\boldsymbol{p}_T$ be the output of the SGM algorithm based on the CLD (Section 2.2) at time $T$, where the data distribution $q$ is the standard Gaussian $\gamma^d$, and the score estimate is exact ($\varepsilon_{\text{score}} = 0$). Suppose that the step size $h$ satisfies $h \lesssim 1/(T \vee 1)$. Then, for the path measures $\boldsymbol{P}_T$ and $\boldsymbol{Q}_T^{\leftarrow}$ of the algorithm and the continuous-time process (7) respectively (see Section C for details), it holds that $\mathsf{KL}(\boldsymbol{Q}_T^{\leftarrow} \parallel \boldsymbol{P}_T) \geq dhT$.*

Theorem 8 shows that in order to make the KL divergence between the path measures small, we must take $h \lesssim 1/d$, which leads to an iteration complexity that scales linearly in the dimension $d$. Theorem 8 is not a proof that SGMs based on the CLD cannot achieve better than linear dimension dependence, as it is possible that the output $\boldsymbol{p}_T$ of the SGM is close to $q \otimes \gamma^d$ even if the path measures are not close, but it rules out the possibility of obtaining a better dimension dependence via our Girsanov proof technique. We believe that it provides compelling evidence for our conjecture, i.e., that under our assumptions, the CLD does not improve the complexity of SGMs over DDPM.

We remark that in this section, we have only considered the error arising from discretization of the SDE. It is possible that the score function $\nabla_v \ln \boldsymbol{q}_t$ for the SGM with the CLD is easier to estimate than the score function for DDPM, providing a *statistical* benefit of using the CLD. Indeed, under the manifold hypothesis, the score $\nabla \ln q_t$ for DDPM blows up at $t = 0$, but the score $\nabla_v \ln \boldsymbol{q}_t$ for CLD is well-defined at $t = 0$, and hence may lead to improvements over DDPM. We do not investigate this question here and leave it as future work.

## 4 TECHNICAL OVERVIEW

We now give a detailed technical overview for the proof for DDPM (Theorem 2). The proof for CLD (Theorem 7) follows along similar lines.

Recall that we must deal with three sources of error: (1) the estimation of the score function; (2) the discretization of the SDE; and (3) the initialization of the reverse process at $\gamma^d$ rather than at $q_T$.

First, we ignore the errors (1) and (2), and focus on the error (3). Hence, we consider the continuous-time reverse SDE (4), initialized from $\gamma^d$ (resp. $q_T$) and denote by $\tilde{p}_t$ (resp. $q_{T-t}$)

its marginal distributions. Note that $\tilde{p}_0 = \gamma^d$ and that $q_0 = q$, the data distribution. First, using the exponential contraction of the KL divergence along the (forward) OU process, we have $\mathsf{KL}(q_T \| \gamma^d) \le \exp(-2T)\,\mathsf{KL}(q \| \gamma^d)$. Then, using the data processing inequality along the backward process, we have $\mathsf{TV}(\tilde{p}_T, q) \le \mathsf{TV}(\gamma^d, q_T)$. Therefore, using Pinsker inequality, we get

$$\mathsf{TV}(\tilde{p}_T, q) \le \mathsf{TV}(\gamma^d, q_T) \le \sqrt{\mathsf{KL}(q_T \| \gamma^d)} \le \exp(-T)\sqrt{\mathsf{KL}(q \| \gamma^d)},$$

i.e., the output $\tilde{p}_T$ converges to the data distribution $q$ exponentially fast as $T \to \infty$.

Next, we consider the score estimation error (1) and the discretization error (2). Using Girsanov's theorem, these errors can be bounded by

$$\sum_{k=0}^{N-1} \mathbb{E}\int_{kh}^{(k+1)h} \|s_{T-kh}(\bar{X}_{kh}^{\leftarrow}) - \nabla \ln q_{T-t}(\bar{X}_t^{\leftarrow})\|^2 \, \mathrm{d}t \tag{8}$$

(see the inequality (15) in the supplement). Unlike other proof techniques, such as the interpolation method in Lee et al. (2022a), the error term (8) in Girsanov's theorem involves an expectation under the law of the true reverse process, instead of the law of the algorithm (see Lee et al. (2022a)). This difference allows us to bound the score estimation error using Assumption 3 directly, which allows a simpler proof that works under milder assumptions on the data distribution. However, the use of Girsanov's theorem typically requires a technical condition known as *Novikov's condition*, which *fails* to hold under under our minimal assumptions. To circumvent this issue, we use an approximation argument relying on abstract results on the convergence of stochastic processes. A recent concurrent and independent work Liu et al. (2022) also uses Girsanov's theorem, but assumes that Novikov's condition holds at the outset.

## 5 CONCLUSION

In this work, we provided the first convergence guarantees for SGMs which hold under realistic assumptions (namely, $L^2$-accurate score estimation and arbitrarily non-log-concave data distributions) and which scale polynomially in the problem parameters. Our results take a step towards explaining the remarkable empirical success of SGMs, at least assuming the score is learned with small $L^2$ error.

The main limitation of this work is that we did not address the question of when the score function can be learned well. In general, studying the non-convex training dynamics of learning the score function via neural networks is challenging, but we believe that the resolution of this problem, even for simple learning tasks, would shed considerable light on SGMs. Together with the results in this paper, it would yield the first end-to-end guarantees for SGMs.

In light of the interpretation of our result as a reduction of the task of sampling to the task of score function estimation, we also ask whether there are interesting situations where it is easier to learn the score function (not necessarily via a neural network) than to (directly) sample.

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

## A PRELIMINARIES

In this section, we review the notion of score matching and provide a list of notation for the proofs.

### A.1 PRIMER ON SCORE MATCHING

In order to estimate the score function $\nabla \ln q_t$, consider minimizing the $L^2(q_t)$ loss over a function class $\mathscr{F}$,

$$\underset{s_t \in \mathscr{F}}{\text{minimize}} \quad \mathbb{E}_{q_t}[\|s_t - \nabla \ln q_t\|^2], \tag{9}$$

where $\mathscr{F}$ could be, e.g., a class of neural networks. The idea of score matching, which goes back to Hyvärinen (2005); Vincent (2011), is that after applying integration by parts for the Gaussian measure, the problem (9) is *equivalent* to the following problem:

$$\underset{s_t \in \mathscr{F}}{\text{minimize}} \quad \mathbb{E}\left[\left\|s_t(\bar{X}_t) + \frac{1}{\sqrt{1 - \exp(-2t)}} Z_t\right\|^2\right], \tag{10}$$

where $Z_t \sim \mathsf{normal}(0, I_d)$ is independent of $\bar{X}_0$ and $\bar{X}_t = \exp(-t)\,\bar{X}_0 + \sqrt{1 - \exp(-2t)}\, Z_t$, in the sense that (9) and (10) share the same minimizers. We give a self-contained derivation in Appendix D for the sake of completeness. Unlike (9), however, the objective in (10) can be replaced with an empirical version and estimated on the basis of samples $\bar{X}_0^{(1)}, \ldots, \bar{X}_0^{(n)}$ from $q$, leading to the finite-sample problem

$$\underset{s_t \in \mathscr{F}}{\text{minimize}} \quad \frac{1}{n} \sum_{i=1}^{n} \left\|s_t(\bar{X}_t^{(i)}) + \frac{1}{\sqrt{1 - \exp(-2t)}} Z_t^{(i)}\right\|^2,$$

where $(Z_t^{(i)})_{i \in [n]}$ are i.i.d. standard Gaussians independent of $(\bar{X}_0^{(i)})_{i \in [n]}$. Moreover, if we parameterize the score as $s_t = -\frac{1}{\sqrt{1-\exp(-2t)}}\,\widehat{z}_t$, then the empirical problem is equivalent to

$$\underset{\widehat{z}_t \in -\sqrt{1-\exp(-2t)}\,\mathscr{F}}{\text{minimize}} \quad \frac{1}{n} \sum_{i=1}^{n} \left\|\widehat{z}_t(\bar{X}_t^{(i)}) - Z_t^{(i)}\right\|^2,$$

which has the illuminating interpretation of predicting the added noise $Z_t^{(i)}$ from the noised data $\bar{X}_t^{(i)}$, i.e., denoising.

### NOTATION

For a measurable mapping $T : \mathsf{X} \to \mathsf{X}$ and a measure $\mu$ on $\mathsf{X}$, where $\mathsf{X}$ is a measurable space, the notation $T_{\#}\mu$ refers to the pushforward of $\mu$ by the mapping $T$, i.e., if $X \sim \mu$, then $T(X) \sim T_{\#}\mu$.

**Stochastic processes and their laws.**

- The data distribution is $q = q_0$.
- The forward process (1) is denoted $(\bar{X}_t)_{t \in [0,T]}$, and $\bar{X}_t \sim q_t$.
- The reverse process (4) is denoted $(\bar{X}_t^{\leftarrow})_{t \in [0,T]}$, where $\bar{X}_t^{\leftarrow} := \bar{X}_{T-t} \sim q_{T-t}$.
- The SGM algorithm (5) is denoted $(X_t^{\leftarrow})_{t \in [0,T]}$, and $X_t^{\leftarrow} \sim p_t$. Recall that we initialize at $p_0 = \gamma^d$, the standard Gaussian measure.
- The process $(X_t^{\leftarrow, q_T})_{t \in [0,T]}$ is the same as $(X_t^{\leftarrow})_{t \in [0,T]}$, except that we initialize this process at $q_T$ rather than at $\gamma^d$. We write $X_t^{\leftarrow, q_T} \sim p_t^{q_T}$.

**Conventions for Girsanov's theorem.** When we apply Girsanov's theorem, it is convenient to instead think about a single stochastic process, which for ease of notation we denote simply via $(X_t)_{t \in [0,T]}$, and we consider different measures over the path space $\mathcal{C}([0,T]; \mathbb{R}^d)$.

The two measures we consider over path space are:

- $Q_T^{\leftarrow}$, under which $(X_t)_{t \in [0,T]}$ has the law of the reverse process (4);
- $P_T^{q_T}$, under which $(X_t)_{t \in [0,T]}$ has the law of the SGM algorithm initialized at $q_T$ (corresponding to the process $(X_t^{\leftarrow, q_T})_{t \in [0,T]}$ defined above).

We also use the following notion from stochastic calculus (Le Gall, 2016, Definition 4.6):

- A local martingale $(\mathcal{L}_t)_{t \in [0,T]}$ is a stochastic process s.t. there exists a sequence of non-decreasing stopping times $T_n \to T$ s.t. $\mathcal{L}^n = (\mathcal{L}_{t \wedge T_n})_{t \in [0,T]}$ is a martingale.

**Other parameters.** We recall that $T > 0$ denotes the total time for which we run the forward process; $h > 0$ is the step size of the discretization; $L \geq 1$ is the Lipschitz constant of the score function; $\mathfrak{m}_2^2 := \mathbb{E}_q[\|\cdot\|^2]$ is the second moment under the data distribution; and $\varepsilon_{\text{score}}$ is the $L^2$ score estimation error.

**Notation for CLD.** The notational conventions for the CLD are similar; however, we must also consider a velocity variable $V$. When discussing quantities which involve both position and velocity (e.g., the joint distribution $\boldsymbol{q}_t$ of $(\bar{X}_t, \bar{V}_t)$), we typically use boldface fonts.

## B  PROOFS FOR DDPM

### B.1  PRELIMINARIES ON GIRSANOV'S THEOREM AND A FIRST ATTEMPT AT APPLYING GIRSANOV'S THEOREM

First, we recall a consequence of Girsanov's theorem that can be obtained by combining Pages 136–139, Theorem 5.22, and Theorem 4.13 of Le Gall (2016).

**Theorem 9.** *For $t \in [0,T]$, let $\mathcal{L}_t = \int_0^t b_s \, \mathrm{d}B_s$ where $B$ is a $Q$-Brownian motion. Assume that $\mathbb{E}_Q \int_0^T \|b_s\|^2 \, \mathrm{d}s < \infty$. Then, $\mathcal{L}$ is a $Q$-martingale in $L^2(Q)$. Moreover, if*

$$\mathbb{E}_Q \, \mathcal{E}(\mathcal{L})_T = 1, \quad \text{where} \quad \mathcal{E}(\mathcal{L})_t := \exp\left(\int_0^t b_s \, \mathrm{d}B_s - \frac{1}{2} \int_0^t \|b_s\|^2 \, \mathrm{d}s\right), \tag{11}$$

*then $\mathcal{E}(\mathcal{L})$ is also a $Q$-martingale and the process*

$$t \mapsto B_t - \int_0^t b_s \mathrm{d}s$$

*is a Brownian motion under $P := \mathcal{E}(\mathcal{L})_T \, Q$, the probability distribution with density $\mathcal{E}(\mathcal{L})_T$ w.r.t. $Q$.*

*If the assumptions of Girsanov's theorem are satisfied* (i.e., the condition (11)), we can apply Girsanov's theorem to $Q = Q_T^{\leftarrow}$ and

$$b_t = \sqrt{2} \left( s_{T-kh}(X_{kh}) - \nabla \ln q_{T-t}(X_t) \right),$$

where $t \in [kh, (k+1)h]$. This tells us that under $P = \mathcal{E}(\mathcal{L})_T \, Q_T^{\leftarrow}$, there exists a Brownian motion $(\beta_t)_{t \in [0,T]}$ s.t.

$$\mathrm{d}B_t = \sqrt{2} \left( s_{T-kh}(X_{kh}) - \nabla \ln q_{T-t}(X_t) \right) \mathrm{d}t + \mathrm{d}\beta_t. \tag{12}$$

Recall that under $Q_T^{\leftarrow}$ we have a.s.

$$\mathrm{d}X_t = \{X_t + 2 \nabla \ln q_{T-t}(X_t)\} \, \mathrm{d}t + \sqrt{2} \, \mathrm{d}B_t, \qquad X_0 \sim q_T. \tag{13}$$

The equation above still holds $P$-a.s. since $P \ll Q_T^{\leftarrow}$ (even if $B$ is no longer a $P$-Brownian motion). Plugging (12) into (13) we have $P$-a.s.,[3]

$$\mathrm{d}X_t = \{X_t + 2 \, s_{T-kh}(X_{kh})\} \, \mathrm{d}t + \sqrt{2} \, \mathrm{d}\beta_t, \qquad X_0 \sim q_T.$$

---

[3]We still have $X_0 \sim q_T$ under $P$ because the marginal at time $t = 0$ of $P$ is equal to the marginal at time $t = 0$ of $Q_T^{\leftarrow}$. That is a consequence of the fact that $\mathcal{E}(\mathcal{L})$ is a (true) $Q_T^{\leftarrow}$-martingale.

In other words, under $P$, the distribution of $X$ is the SGM algorithm started at $q_T$, i.e., $P = P_T^{q_T} = \mathcal{E}(\mathcal{L})_T \, Q_T^{\leftarrow}$. Therefore,

$$\mathsf{KL}(Q_T^{\leftarrow} \parallel P_T^{q_T}) = \mathbb{E}_{Q_T^{\leftarrow}} \ln \frac{\mathrm{d}Q_T^{\leftarrow}}{\mathrm{d}P_T^{q_T}} = \mathbb{E}_{Q_T^{\leftarrow}} \ln \mathcal{E}(\mathcal{L})_T^{-1} \tag{14}$$

$$= \sum_{k=0}^{N-1} \mathbb{E}_{Q_T^{\leftarrow}} \int_{kh}^{(k+1)h} \|s_{T-kh}(X_{kh}) - \nabla \ln q_{T-t}(X_t)\|^2 \, \mathrm{d}t \, ,$$

where we used $\mathbb{E}_{Q_T^{\leftarrow}} \mathcal{L}_t = 0$ because $\mathcal{L}$ is a martingale.

The equality (14) allows us to bound the discrepancy between the SGM algorithm and the reverse process.

## B.2 CHECKING THE ASSUMPTIONS OF GIRSANOV'S THEOREM AND THE GIRSANOV DISCRETIZATION ARGUMENT

In most applications of Girsanov's theorem in sampling, a sufficient condition for (11) to hold, known as *Novikov's condition*, is satisfied. Here, Novikov's condition writes

$$\mathbb{E}_{Q_T^{\leftarrow}} \exp\Big(\sum_{k=0}^{N-1} \int_{kh}^{(k+1)h} \|s_{T-kh}(X_{kh}) - \nabla \ln q_{T-t}(X_t)\|^2 \, \mathrm{d}t\Big) < \infty \, ,$$

and if Novikov's condition holds, we can apply Girsanov's theorem directly. However, under Assumptions 1, 2, and 3 alone, Novikov's condition need not hold. Indeed, in order to check Novikov's condition, we would want $X_0$ to have sub-Gaussian tails for instance.

Furthermore, we also could not check that the condition (11), which is weaker than Novikov's condition, holds. Therefore, in the proof of the next Theorem, we use a approximation technique to show that

$$\mathsf{KL}(Q_T^{\leftarrow} \parallel P_T^{q_T}) = \mathbb{E}_{Q_T^{\leftarrow}} \ln \frac{\mathrm{d}Q_T^{\leftarrow}}{\mathrm{d}P_T^{q_T}} \leq \mathbb{E}_{Q_T^{\leftarrow}} \ln \mathcal{E}(\mathcal{L})_T^{-1} \tag{15}$$

$$= \sum_{k=0}^{N-1} \mathbb{E}_{Q_T^{\leftarrow}} \int_{kh}^{(k+1)h} \|s_{T-kh}(X_{kh}) - \nabla \ln q_{T-t}(X_t)\|^2 \, \mathrm{d}t \, .$$

We then use a discretization argument based on stochastic calculus to further bound this quantity. The result is the following theorem.

**Theorem 10** (discretization error for DDPM). *Suppose that Assumptions 1, 2, and 3 hold. Let $Q_T^{\leftarrow}$ and $P_T^{q_T}$ denote the measures on path space corresponding to the reverse process* (4) *and the SGM algorithm with $L^2$-accurate score estimate initialized at $q_T$. Assume that $L \geq 1$ and $h \lesssim 1/L$. Then,*

$$\mathsf{TV}(P_T^{q_T}, Q_T^{\leftarrow})^2 \leq \mathsf{KL}(Q_T^{\leftarrow} \parallel P_T^{q_T}) \lesssim (\varepsilon_{\mathrm{score}}^2 + L^2 dh + L^2 \mathfrak{m}_2^2 h^2) \, T \, .$$

*Proof.* We start by proving

$$\sum_{k=0}^{N-1} \mathbb{E}_{Q_T^{\leftarrow}} \int_{kh}^{(k+1)h} \|s_{T-kh}(X_{kh}) - \nabla \ln q_{T-t}(X_t)\|^2 \, \mathrm{d}t \lesssim (\varepsilon_{\mathrm{score}}^2 + L^2 dh + L^2 \mathfrak{m}_2^2 h^2) \, T \, .$$

Then, we give the approximation argument to prove the inequality (15).

**Bound on the discretization error.** For $t \in [kh, (k+1)h]$, we can decompose

$$\mathbb{E}_{Q_T^{\leftarrow}}[\|s_{T-kh}(X_{kh}) - \nabla \ln q_{T-t}(X_t)\|^2]$$

$$\lesssim \mathbb{E}_{Q_T^{\leftarrow}}[\|s_{T-kh}(X_{kh}) - \nabla \ln q_{T-kh}(X_{kh})\|^2]$$

$$+ \mathbb{E}_{Q_T^{\leftarrow}}[\|\nabla \ln q_{T-kh}(X_{kh}) - \nabla \ln q_{T-t}(X_{kh})\|^2]$$

$$+ \mathbb{E}_{Q_T^{\leftarrow}}[\|\nabla \ln q_{T-t}(X_{kh}) - \nabla \ln q_{T-t}(X_t)\|^2]$$

$$\lesssim \varepsilon_{\mathrm{score}}^2 + \mathbb{E}_{Q_T^{\leftarrow}}\Big[\Big\|\nabla \ln \frac{q_{T-kh}}{q_{T-t}}(X_{kh})\Big\|^2\Big] + L^2 \, \mathbb{E}_{Q_T^{\leftarrow}}[\|X_{kh} - X_t\|^2] \, . \tag{16}$$

We must bound the change in the score function along the forward process. If $S : \mathbb{R}^d \to \mathbb{R}^d$ is the mapping $S(x) := \exp(-(t - kh)) x$, then $q_{T-kh} = S_\# q_{T-t} * \mathsf{normal}(0, 1 - \exp(-2(t - kh)))$. We can then use Lee et al. (2022a, Lemma C.12) (or the more general Lemma 17 that we prove in Section C.4) with $\alpha = \exp(t - kh) = 1 + O(h)$ and $\sigma^2 = 1 - \exp(-2(t - kh)) = O(h)$ to obtain

$$
\left\| \nabla \ln \frac{q_{T-kh}}{q_{T-t}}(X_{kh}) \right\|^2 \lesssim L^2 dh + L^2 h^2 \|X_{kh}\|^2 + (1 + L^2) h^2 \|\nabla \ln q_{T-t}(X_{kh})\|^2
$$
$$
\lesssim L^2 dh + L^2 h^2 \|X_{kh}\|^2 + L^2 h^2 \|\nabla \ln q_{T-t}(X_{kh})\|^2
$$

where the last line uses $L \geq 1$.

For the last term,

$$
\|\nabla \ln q_{T-t}(X_{kh})\|^2 \lesssim \|\nabla \ln q_{T-t}(X_t)\|^2 + \|\nabla \ln q_{T-t}(X_{kh}) - \nabla \ln q_{T-t}(X_t)\|^2
$$
$$
\lesssim \|\nabla \ln q_{T-t}(X_t)\|^2 + L^2 \|X_{kh} - X_t\|^2 ,
$$

where the second term above is absorbed into the third term of the decomposition (16). Hence,

$$
\mathbb{E}_{Q_T^\leftarrow}[\|s_{T-kh}(X_{kh}) - \nabla \ln q_{T-t}(X_t)\|^2]
$$
$$
\lesssim \varepsilon_{\text{score}}^2 + L^2 dh + L^2 h^2 \, \mathbb{E}_{Q_T^\leftarrow}[\|X_{kh}\|^2]
$$
$$
+ L^2 h^2 \, \mathbb{E}_{Q_T^\leftarrow}[\|\nabla \ln q_{T-t}(X_t)\|^2] + L^2 \, \mathbb{E}_{Q_T^\leftarrow}[\|X_{kh} - X_t\|^2] .
$$

Using the fact that under $Q_T^\leftarrow$, the process $(X_t)_{t \in [0,T]}$ is the time reversal of the forward process $(\bar{X}_t)_{t \in [0,T]}$, we can apply the moment bounds in Lemma 11 and the movement bound in Lemma 12 to obtain

$$
\mathbb{E}_{Q_T^\leftarrow}[\|s_{T-kh}(X_{kh}) - \nabla \ln q_{T-t}(X_t)\|^2]
$$
$$
\lesssim \varepsilon_{\text{score}}^2 + L^2 dh + L^2 h^2 (d + \mathfrak{m}_2^2) + L^3 dh^2 + L^2 (\mathfrak{m}_2^2 h^2 + dh)
$$
$$
\lesssim \varepsilon_{\text{score}}^2 + L^2 dh + L^2 \mathfrak{m}_2^2 h^2 .
$$

**Approximation argument.** For $t \in [0, T]$, let $\mathcal{L}_t = \int_0^t b_s \, \mathrm{d}B_s$ where $B$ is a $Q_T^\leftarrow$-Brownian motion and we define
$$
b_t = \sqrt{2} \{s_{T-kh}(X_{kh}) - \nabla \ln q_{T-t}(X_t)\} ,
$$
for $t \in [kh, (k+1)h]$. We proved that $\mathbb{E}_{Q_T^\leftarrow} \int_0^T \|b_s\|^2 \, \mathrm{d}s \lesssim (\varepsilon_{\text{score}}^2 + L^2 dh + L^2 \mathfrak{m}_2^2 h^2) T < \infty$. Using Le Gall (2016, Proposition 5.11), $(\mathcal{E}(\mathcal{L})_t)_{t \in [0,T]}$ is a local martingale. Therefore, there exists a non-decreasing sequence of stopping times $T_n \nearrow T$ s.t. $(\mathcal{E}(\mathcal{L})_{t \wedge T_n})_{t \in [0,t]}$ is a martingale. Note that $\mathcal{E}(\mathcal{L})_{t \wedge T_n} = \mathcal{E}(\mathcal{L}^n)_t$ where $\mathcal{L}_t^n = \mathcal{L}_{t \wedge T_n}$. Since $\mathcal{E}(\mathcal{L}^n)$ is a martingale, we have

$$
\mathbb{E}_{Q_T^\leftarrow} \mathcal{E}(\mathcal{L}^n)_T = \mathbb{E}_{Q_T^\leftarrow} \mathcal{E}(\mathcal{L}^n)_0 = 1 ,
$$

i.e., $\mathbb{E}_{Q_T^\leftarrow} \mathcal{E}(\mathcal{L})_{T_n} = 1$.

We apply Girsanov's theorem to $\mathcal{L}_t^n = \int_0^t b_s \, \mathbb{1}_{[0,T_n]}(s) \, \mathrm{d}B_s$, where $B$ is a $Q_T^\leftarrow$-Brownian motion. Since $\mathbb{E}_{Q_T^\leftarrow} \int_0^T \|b_s \mathbb{1}_{[0,T_n]}(s)\|^2 \, \mathrm{d}s \leq \mathbb{E}_{Q_T^\leftarrow} \int_0^T \|b_s\|^2 \, \mathrm{d}s < \infty$ (see the last paragraph) and $\mathbb{E}_{Q_T^\leftarrow} \mathcal{E}(\mathcal{L}^n)_T = 1$, we obtain that under $P^n := \mathcal{E}(\mathcal{L}^n)_T \, Q_T^\leftarrow$ there exists a Brownian motion $\beta^n$ s.t. for $t \in [0, T]$,

$$
\mathrm{d}B_t = \sqrt{2} \{s_{T-kh}(X_{kh}) - \nabla \ln q_{T-t}(X_t)\} \mathbb{1}_{[0,T_n]}(t) \, \mathrm{d}t + \mathrm{d}\beta_t^n .
$$

Recall that under $Q_T^\leftarrow$ we have a.s.

$$
\mathrm{d}X_t = \{X_t + 2 \nabla \ln q_{T-t}(X_t)\} \, \mathrm{d}t + \sqrt{2} \, \mathrm{d}B_t , \qquad X_0 \sim q_T .
$$

The equation above still holds $P^n$-a.s. since $P^n \ll Q_T^\leftarrow$. Combining the last two equations we then obtain $P^n$-a.s.,

$$
\mathrm{d}X_t = \{X_t + 2\, s_{T-kh}(X_{kh})\} \mathbb{1}_{[0,T_n]}(t) \, \mathrm{d}t + \{X_t + 2 \nabla \ln q_{T-t}(X_t)\} \mathbb{1}_{[T_n,T]}(t) \, \mathrm{d}t + \sqrt{2} \, \mathrm{d}\beta_t^n , \quad (17)
$$

and $X_0 \sim q_T$. In other words, $P^n$ is the law of the solution of the SDE (17). At this stage we have the bound

$$
\mathsf{KL}(Q_T^{\leftarrow} \parallel P^n) = \mathbb{E}_{Q_T^{\leftarrow}} \ln \mathcal{E}(\mathcal{L})_{T_n}^{-1} = \mathbb{E}_{Q_T^{\leftarrow}} \left[ -\mathcal{L}_{T_n} + \frac{1}{2} \int_0^{T_n} \|b_s\|^2 \, \mathrm{d}s \right] = \mathbb{E}_{Q_T^{\leftarrow}} \frac{1}{2} \int_0^{T_n} \|b_s\|^2 \, \mathrm{d}s
$$

$$
\leq \mathbb{E}_{Q_T^{\leftarrow}} \frac{1}{2} \int_0^T \|b_s\|^2 \, \mathrm{d}s \lesssim (\varepsilon_{\mathrm{score}}^2 + L^2 dh + L^2 \mathfrak{m}_2^2 h^2) \, T,
$$

where we used that $\mathbb{E}_{Q_T^{\leftarrow}} \mathcal{L}_{T_n} = 0$ because $\mathcal{L}$ is a $Q_T^{\leftarrow}$-martingale and $T_n$ is a bounded stopping time (Le Gall, 2016, Corollary 3.23). Our goal is now to show that we can obtain the final result by an approximation argument.

We consider a coupling of $(P^n)_{n \in \mathbb{N}}, P_T^{q_T}$: a sequence of stochastic processes $(X^n)_{n \in \mathbb{N}}$ over the same probability space, a stochastic process $X$ and a single Brownian motion $W$ over that space s.t.[4]

$$
\mathrm{d}X_t^n = \{X_t^n + 2\, s_{T-kh}(X_{kh}^n)\} \mathbb{1}_{[0,T_n]}(t) \, \mathrm{d}t + \{X_t^n + 2 \nabla \ln q_{T-t}(X_t^n)\} \mathbb{1}_{[T_n,T]}(t) \, \mathrm{d}t + \sqrt{2} \, \mathrm{d}W_t \,,
$$

and

$$
\mathrm{d}X_t = \{X_t + 2\, s_{T-kh}(X_{kh}^n)\} \, \mathrm{d}t + \sqrt{2} \, \mathrm{d}W_t \,,
$$

with $X_0 = X_0^n$ a.s. and $X_0 \sim q_T$. Note that the distribution of $X^n$ (resp. $X$) is $P^n$ (resp. $P_T^{q_T}$).

Let $\varepsilon > 0$ and consider the map $\pi_\varepsilon : \mathcal{C}([0,T]; \mathbb{R}^d) \to \mathcal{C}([0,T]; \mathbb{R}^d)$ defined by

$$
\pi_\varepsilon(\omega)(t) := \omega(t \wedge T - \varepsilon) \,.
$$

Noting that $X_t^n = X_t$ for every $t \in [0, T_n]$ and using Lemma 13, we have $\pi_\varepsilon(X^n) \to \pi_\varepsilon(X)$ a.s., uniformly over $[0,T]$. Therefore, $\pi_{\varepsilon\#} P^n \to \pi_{\varepsilon\#} P_T^{q_T}$ weakly. Using the lower semicontinuity of the KL divergence and the data-processing inequality (Ambrosio et al., 2005, Lemma 9.4.3 and Lemma 9.4.5), we obtain

$$
\mathsf{KL}((\pi_\varepsilon)_\# Q_T^{\leftarrow} \parallel (\pi_\varepsilon)_\# P_T^{q_T}) \leq \liminf_{n \to \infty} \mathsf{KL}((\pi_\varepsilon)_\# Q_T^{\leftarrow} \parallel (\pi_\varepsilon)_\# P^n)
$$

$$
\leq \liminf_{n \to \infty} \mathsf{KL}(Q_T^{\leftarrow} \parallel P^n)
$$

$$
\lesssim (\varepsilon_{\mathrm{score}}^2 + L^2 dh + L^2 \mathfrak{m}_2^2 h^2) \, T \,.
$$

Finally, using Lemma 14, $\pi_\varepsilon(\omega) \to \omega$ as $\varepsilon \to 0$, uniformly over $[0,T]$. Therefore, using Ambrosio et al. (2005, Corollary 9.4.6), $\mathsf{KL}((\pi_\varepsilon)_\# Q_T^{\leftarrow} \parallel (\pi_\varepsilon)_\# P_T^{q_T}) \to \mathsf{KL}(Q_T^{\leftarrow} \parallel P_T^{q_T})$ as $\varepsilon \searrow 0$. Therefore,

$$
\mathsf{KL}(Q_T^{\leftarrow} \parallel P_T^{q_T}) \lesssim (\varepsilon_{\mathrm{score}}^2 + L^2 dh + L^2 \mathfrak{m}_2^2 h^2) \, T \,.
$$

We conclude with Pinsker's inequality ($\mathsf{TV}^2 \leq \mathsf{KL}$). $\qquad\square$

### B.3 Proof of Theorem 2

We can now conclude our main result.

*Proof of Theorem 2.* We recall the notation from Section 4. By the data processing inequality,

$$
\mathsf{TV}(p_T, q) \leq \mathsf{TV}(P_T, P_T^{q_T}) + \mathsf{TV}(P_T^{q_T}, Q_T^{\leftarrow}) \leq \mathsf{TV}(q_T, \gamma^d) + \mathsf{TV}(P_T^{q_T}, Q_T^{\leftarrow}) \,.
$$

Using the convergence of the OU process in KL divergence (see, e.g., Bakry et al., 2014, Theorem 5.2.1) and applying Theorem 10 for the second term,

$$
\mathsf{TV}(p_T, q) \lesssim \sqrt{\mathsf{KL}(q \parallel \gamma^d)} \exp(-T) + (\varepsilon_{\mathrm{score}} + L\sqrt{dh} + L\mathfrak{m}_2 h) \sqrt{T} \,,
$$

which proves the result. $\qquad\square$

---

[4]Such a coupling always exists, see Le Gall (2016, Corollary 8.5).

### B.4 Auxiliary lemmas

In this section, we prove some auxiliary lemmas which are used in the proof of Theorem 2.

**Lemma 11** (moment bounds for DDPM). *Suppose that Assumptions 1 and 2 hold. Let $(\bar{X}_t)_{t\in[0,T]}$ denote the forward process* (1).

1. *(moment bound) For all $t \geq 0$,*

$$\mathbb{E}[\|\bar{X}_t\|^2] \leq d \vee \mathfrak{m}_2^2\,.$$

2. *(score function bound) For all $t \geq 0$,*

$$\mathbb{E}[\|\nabla \ln q_t(\bar{X}_t)\|^2] \leq Ld\,.$$

*Proof.* 1. Along the OU process, we have $\bar{X}_t \stackrel{\mathsf{d}}{=} \exp(-t)\,\bar{X}_0 + \sqrt{1 - \exp(-2t)}\,\xi$, where $\xi \sim \mathsf{normal}(0, I_d)$ is independent of $\bar{X}_0$. Hence,

$$\mathbb{E}[\|\bar{X}_t\|^2] = \exp(-2t)\,\mathbb{E}[\|X\|^2] + \{1 - \exp(-2t)\}\,d \leq d \vee \mathfrak{m}_2^2\,. \qquad \square$$

2. This follows from the $L$-smoothness of $\ln q_t$ (see, e.g., Vempala & Wibisono, 2019, Lemma 9). We give a short proof for the sake of completeness.

If $\mathscr{L}_t f := \Delta f - \langle \nabla U_t, \nabla f \rangle$ is the generator associated with $q_t \propto \exp(-U_t)$, then

$$0 = \mathbb{E}_{q_t}\,\mathscr{L}U_t = \mathbb{E}_{q_t}\,\Delta U_t - \mathbb{E}_{q_t}[\|\nabla U_t\|^2] \leq Ld - \mathbb{E}_{q_t}[\|\nabla U_t\|^2]\,.$$

**Lemma 12** (movement bound for DDPM). *Suppose that Assumption 2 holds. Let $(\bar{X}_t)_{t\in[0,T]}$ denote the forward process* (1). *For $0 \leq s < t$ with $\delta := t - s$, if $\delta \leq 1$, then*

$$\mathbb{E}[\|\bar{X}_t - \bar{X}_s\|^2] \lesssim \delta^2\mathfrak{m}_2^2 + \delta d\,.$$

*Proof.* We can write

$$\begin{aligned}
\mathbb{E}[\|\bar{X}_t - \bar{X}_s\|^2] &= \mathbb{E}\Big[\Big\|-\int_s^t \bar{X}_r\,\mathrm{d}r + \sqrt{2}\,(B_t - B_s)\Big\|^2\Big] \\
&\lesssim \delta \int_s^t \mathbb{E}[\|\bar{X}_r\|^2]\,\mathrm{d}r + \delta d \lesssim \delta^2\,(d + \mathfrak{m}_2^2) + \delta d \\
&\lesssim \delta^2\mathfrak{m}_2^2 + \delta d\,,
\end{aligned}$$

where we used Lemma 11. $\qquad \square$

We omit the proofs of the two next lemmas as they are straightforward.

**Lemma 13.** *Consider $f_n, f : [0,T] \to \mathbb{R}^d$ s.t. there exists an increasing sequence $(T_n)_{n\in\mathbb{N}} \subseteq [0,T]$ satisfying the conditions*

- *$T_n \to T$ as $n \to \infty$,*

- *$f_n(t) = f(t)$ for every $t \leq T_n$.*

*Then, for every $\varepsilon > 0$, $f_n \to f$ uniformly over $[0, T - \varepsilon]$. In particular, $f_n(\cdot \wedge T - \varepsilon) \to f(\cdot \wedge T - \varepsilon)$ uniformly over $[0,T]$.*

**Lemma 14.** *Consider $f : [0,T] \to \mathbb{R}^d$ continuous, and $f_\varepsilon : [0,T] \to \mathbb{R}^d$ s.t. $f_\varepsilon(t) = f(t \wedge (T - \varepsilon))$ for $\varepsilon > 0$. Then $f_\varepsilon \to f$ uniformly over $[0,T]$ as $\varepsilon \to 0$.*

## B.5 PROOF OF COROLLARY 6

*Proof of Corollary 6.* For $R_0 > 0$, let $\Pi_{R_0}$ denote the projection onto $\mathsf{B}(0, R_0)$. We want to prove that $W_2((\Pi_{R_0})_\# p_{T-t}, q) \leq \varepsilon$. We use the decomposition

$$W_2((\Pi_{R_0})_\# p_{T-t}, q) \leq W_2((\Pi_{R_0})_\# p_{T-t}, (\Pi_{R_0})_\# q_t) + W_2((\Pi_{R_0})_\# q_t, q).$$

For the first term, since $(\Pi_{R_0})_\# p_{T-t}$ and $(\Pi_{R_0})_\# q_t$ both have support contained in $\mathsf{B}(0, R_0)$, we can upper bound the Wasserstein distance by the total variation distance. Namely, Rolland (2022, Lemma 9) implies that

$$W_2((\Pi_{R_0})_\# p_{T-t}, (\Pi_{R_0})_\# q_t) \lesssim R_0 \sqrt{\mathsf{TV}((\Pi_{R_0})_\# p_{T-t}, (\Pi_{R_0})_\# q_t)} + R_0 \exp(-R_0).$$

By the data-processing inequality,

$$\mathsf{TV}((\Pi_{R_0})_\# p_{T-t}, (\Pi_{R_0})_\# q_t) \leq \mathsf{TV}(p_{T-t}, q_t) \leq \varepsilon_{\mathrm{TV}},$$

where $\varepsilon_{\mathrm{TV}}$ is from Corollary 4, yielding

$$W_2((\Pi_{R_0})_\# p_{T-t}, (\Pi_{R_0})_\# q_t) \lesssim R_0 \sqrt{\varepsilon_{\mathrm{TV}}} + R_0 \exp(-R_0).$$

Next, we take $R_0 \geq R$ so that $(\Pi_{R_0})_\# q = q$. Since $\Pi_{R_0}$ is 1-Lipschitz, we have

$$W_2((\Pi_{R_0})_\# q_t, q) = W_2((\Pi_{R_0})_\# q_t, (\Pi_{R_0})_\# q) \leq W_2(q_t, q) \leq \varepsilon_{W_2},$$

where $\varepsilon_{W_2}$ is from Corollary 4. Combining these bounds,

$$W_2((\Pi_{R_0})_\# p_{T-t}, q) \lesssim R_0 \sqrt{\varepsilon_{\mathrm{TV}}} + R_0 \exp(-R_0) + \varepsilon_{W_2}.$$

We now take $\varepsilon_{W_2} = \varepsilon/3$, $R_0 = \widetilde{\Theta}(R)$, and $\varepsilon_{\mathrm{TV}} = \widetilde{\Theta}(\varepsilon^2/R^2)$ to obtain the desired result. The iteration complexity follows from Corollary 4. □

## C PROOFS FOR CLD

### C.1 BACKGROUND ON THE CLD PROCESS

More generally, for the forward process we can introduce a *friction parameter* $\gamma > 0$ and consider

$$\mathrm{d}\bar{X}_t = \bar{V}_t \, \mathrm{d}t,$$
$$\mathrm{d}\bar{V}_t = -\bar{X}_t \, \mathrm{d}t - \gamma \bar{V}_t \, \mathrm{d}t + \sqrt{2\gamma} \, \mathrm{d}B_t.$$

If we write $\bar{\boldsymbol{\theta}}_t := (\bar{X}_t, \bar{V}_t)$, then the forward process satisfies the linear SDE

$$\mathrm{d}\bar{\boldsymbol{\theta}}_t = \boldsymbol{A}_\gamma \bar{\boldsymbol{\theta}}_t \, \mathrm{d}t + \Sigma_\gamma \, \mathrm{d}B_t, \qquad \text{where } \boldsymbol{A}_\gamma := \begin{bmatrix} 0 & 1 \\ -1 & -\gamma \end{bmatrix} \text{ and } \Sigma_\gamma := \begin{bmatrix} 0 \\ \sqrt{2\gamma} \end{bmatrix}.$$

The solution to the SDE is given by

$$\bar{\boldsymbol{\theta}}_t = \exp(t\boldsymbol{A}_\gamma) \, \bar{\boldsymbol{\theta}}_0 + \int_0^t \exp\{(t-s)\,\boldsymbol{A}_\gamma\} \Sigma_\gamma \, \mathrm{d}B_s, \tag{18}$$

which means that by the Itô isometry,

$$\mathrm{law}(\bar{\boldsymbol{\theta}}_t) = \exp(t\boldsymbol{A}_\gamma)_\# \, \mathrm{law}(\bar{\boldsymbol{\theta}}_0) * \mathsf{normal}\Big(0, \int_0^t \exp\{(t-s)\,\boldsymbol{A}_\gamma\} \Sigma_\gamma \Sigma_\gamma^\mathsf{T} \exp\{(t-s)\,\boldsymbol{A}_\gamma^\mathsf{T}\} \, \mathrm{d}s\Big).$$

Since $\det \boldsymbol{A}_\gamma = 1$, $\boldsymbol{A}_\gamma$ is always invertible. Moreover, from $\mathrm{tr}\,\boldsymbol{A}_\gamma = -\gamma$, one can work out that the spectrum of $\boldsymbol{A}_\gamma$ is

$$\mathrm{spec}(\boldsymbol{A}_\gamma) = \Big\{ -\frac{\gamma}{2} \pm \sqrt{\frac{\gamma^2}{4} - 1} \Big\}.$$

However, $\boldsymbol{A}_\gamma$ is not diagonalizable. The case of $\gamma = 2$ is special, as it corresponds to the case when the spectrum is $\{-1\}$, and it corresponds to the *critically damped case*. Following Dockhorn et al. (2022), which advocated for setting $\gamma = 2$, we will also only consider the critically damped case. This also has the advantage of substantially simplifying the calculations.

## C.2 GIRSANOV DISCRETIZATION ARGUMENT

In order to apply Girsanov's theorem, we introduce the path measures $\boldsymbol{P}_T^{q_T}$ and $\boldsymbol{Q}_T^{\leftarrow}$, under which

$$
\begin{aligned}
\mathrm{d}X_t &= -V_t \, \mathrm{d}t \,, \\
\mathrm{d}V_t &= \{X_t + 2\,V_t + 4\,\boldsymbol{s}_{T-kh}(X_{kh}, V_{kh})\} \, \mathrm{d}t + 2\,\mathrm{d}B_t \,,
\end{aligned}
$$

for $t \in [kh, (k+1)h]$, and

$$
\begin{aligned}
\mathrm{d}X_t &= -V_t \, \mathrm{d}t \,, \\
\mathrm{d}V_t &= \{X_t + 2\,V_t + 4\,\nabla_v \ln \boldsymbol{q}_{T-t}(X_t, V_t)\} \, \mathrm{d}t + 2\,\mathrm{d}B_t \,,
\end{aligned}
$$

respectively.

Applying Girsanov's theorem, we have the following theorem.

**Corollary 15.** *Suppose that Novikov's condition holds:*

$$
\mathbb{E}_{\boldsymbol{Q}_T^{\leftarrow}} \exp\Big( 2 \sum_{k=0}^{N-1} \int_{kh}^{(k+1)h} \|\boldsymbol{s}_{T-kh}(X_{kh}, V_{kh}) - \nabla_v \ln \boldsymbol{q}_{T-t}(X_t, V_t)\|^2 \, \mathrm{d}t \Big) < \infty \,.
$$

*Then,*

$$
\begin{aligned}
\mathsf{KL}(\boldsymbol{Q}_T^{\leftarrow} \parallel \boldsymbol{P}_T^{q_T}) &= \mathbb{E}_{\boldsymbol{Q}_T^{\leftarrow}} \ln \frac{\mathrm{d}\boldsymbol{Q}_T^{\leftarrow}}{\mathrm{d}\boldsymbol{P}_T^{q_T}} \\
&= 2 \sum_{k=0}^{N-1} \mathbb{E}_{\boldsymbol{Q}_T^{\leftarrow}} \int_{kh}^{(k+1)h} \|\boldsymbol{s}_{T-kh}(X_{kh}, V_{kh}) - \nabla_v \ln \boldsymbol{q}_{T-t}(X_t, V_t)\|^2 \, \mathrm{d}t \,.
\end{aligned}
$$

Similarly to Appendix B.2, even if Novikov's condition does not hold, one can use an approximation to argue that the KL divergence is still upper bounded by the last expression. Since the argument follows along the same lines, we omit it for brevity.

Using this, we now aim to prove the following theorem.

**Theorem 16** (discretization error for CLD). *Suppose that Assumptions 2, 4, and 5 hold. Let $\boldsymbol{Q}_T^{\leftarrow}$ and $\boldsymbol{P}_T^{q_T}$ denote the measures on path space corresponding to the reverse process* (7) *and the SGM algorithm with $L^2$-accurate score estimate initialized at $\boldsymbol{q}_T$. Assume that $L \geq 1$ and $h \lesssim 1/L$. Then,*

$$
\mathsf{TV}(\boldsymbol{P}_T^{q_T}, \boldsymbol{Q}_T^{\leftarrow})^2 \leq \mathsf{KL}(\boldsymbol{Q}_T^{\leftarrow} \parallel \boldsymbol{P}_T^{q_T}) \lesssim (\varepsilon_{\mathrm{score}}^2 + L^2 dh + L^2 \mathfrak{m}_2^2 h^2)\,T \,.
$$

*Proof.* For $t \in [kh, (k+1)h]$, we can decompose

$$
\begin{aligned}
\mathbb{E}_{\boldsymbol{Q}_T^{\leftarrow}} [\|\mathbf{s}_{T-kh}&(X_{kh}, V_{kh}) - \nabla_v \ln \boldsymbol{q}_{T-t}(X_t, V_t)\|^2] \\
&\lesssim \mathbb{E}_{\boldsymbol{Q}_T^{\leftarrow}} [\|\mathbf{s}_{T-kh}(X_{kh}, V_{kh}) - \nabla_v \ln \boldsymbol{q}_{T-kh}(X_{kh}, V_{kh})\|^2] \\
&\quad + \mathbb{E}_{\boldsymbol{Q}_T^{\leftarrow}} [\|\nabla_v \ln \boldsymbol{q}_{T-kh}(X_{kh}, V_{kh}) - \nabla_v \ln \boldsymbol{q}_{T-t}(X_{kh}, V_{kh})\|^2] \\
&\quad + \mathbb{E}_{\boldsymbol{Q}_T^{\leftarrow}} [\|\nabla_v \ln \boldsymbol{q}_{T-t}(X_{kh}, V_{kh}) - \nabla_v \ln \boldsymbol{q}_{T-t}(X_t, V_t)\|^2] \\
&\lesssim \varepsilon_{\mathrm{score}}^2 + \mathbb{E}_{\boldsymbol{Q}_T^{\leftarrow}} \Big[ \Big\| \nabla_v \ln \frac{\boldsymbol{q}_{T-kh}}{\boldsymbol{q}_{T-t}}(X_{kh}, V_{kh}) \Big\|^2 \Big] + L^2 \, \mathbb{E}_{\boldsymbol{Q}_T^{\leftarrow}} [\|(X_{kh}, V_{kh}) - (X_t, V_t)\|^2] \quad (19)
\end{aligned}
$$

The change in the score function is bounded by Lemma 17, which generalizes Lee et al. (2022a, Lemma C.12). From the representation (18) of the solution to the CLD, we note that

$$
\boldsymbol{q}_{T-kh} = (\boldsymbol{M}_0)_\# \boldsymbol{q}_{T-t} * \mathsf{normal}(0, \boldsymbol{M}_1)
$$

with

$$
\begin{aligned}
\boldsymbol{M}_0 &= \exp\big((t - kh)\,\boldsymbol{A}_2\big) \,, \\
\boldsymbol{M}_1 &= \int_0^{t-kh} \exp\{(t - kh - s)\,\boldsymbol{A}_2\} \Sigma_2 \Sigma_2^\mathsf{T} \exp\{(t - kh - s)\,\boldsymbol{A}_2^\mathsf{T}\} \, \mathrm{d}s \,.
\end{aligned}
$$

In particular, since $\|\boldsymbol{A}_2\|_{\mathrm{op}} \lesssim 1$, $\|\boldsymbol{A}_2^{-1}\|_{\mathrm{op}} \lesssim 1$, and $\|\Sigma_2\|_{\mathrm{op}} \lesssim 1$ it follows that $\|\boldsymbol{M}_0\|_{\mathrm{op}} = 1 + O(h)$ and $\|\boldsymbol{M}_1\|_{\mathrm{op}} = O(h)$. Substituting this into Lemma 17, we deduce that if $h \lesssim 1/L$, then

$$
\begin{aligned}
\left\| \nabla_v \ln \frac{\boldsymbol{q}_{T-kh}}{\boldsymbol{q}_{T-t}}(X_{kh}, V_{kh}) \right\|^2 &\leq \left\| \nabla \ln \frac{\boldsymbol{q}_{T-kh}}{\boldsymbol{q}_{T-t}}(X_{kh}, V_{kh}) \right\|^2 \\
&\lesssim L^2 dh + L^2 h^2 \left( \|X_{kh}\|^2 + \|V_{kh}\|^2 \right) + (1 + L^2) h^2 \left\| \nabla \ln \boldsymbol{q}_{T-t}(X_{kh}, V_{kh}) \right\|^2 \\
&\lesssim L^2 dh + L^2 h^2 \left( \|X_{kh}\|^2 + \|V_{kh}\|^2 \right) + L^2 h^2 \left\| \nabla \ln \boldsymbol{q}_{T-t}(X_{kh}, V_{kh}) \right\|^2 ,
\end{aligned}
$$

where in the last step we used $L \geq 1$.

For the last term,

$$
\left\| \nabla \ln \boldsymbol{q}_{T-t}(X_{kh}, V_{kh}) \right\|^2 \lesssim \left\| \nabla \ln \boldsymbol{q}_{T-t}(X_t, V_t) \right\|^2 + L^2 \left\| (X_{kh}, V_{kh}) - (X_t, V_t) \right\|^2 ,
$$

where the second term above is absorbed into the third term of the decomposition (19). Hence,

$$
\begin{aligned}
\mathbb{E}_{\boldsymbol{Q}_T^{\leftarrow}} [ \| \mathbf{s}_{T-kh}(X_{kh}, V_{kh}) - \nabla_v \ln \boldsymbol{q}_{T-t}(X_t, V_t) \|^2 ] \\
\lesssim \varepsilon_{\mathrm{score}}^2 + L^2 dh + L^2 h^2 \, \mathbb{E}_{\boldsymbol{Q}_T^{\leftarrow}} [ \|X_{kh}\|^2 + \|V_{kh}\|^2 ] \\
+ L^2 h^2 \, \mathbb{E}_{\boldsymbol{Q}_T^{\leftarrow}} [ \| \nabla \ln \boldsymbol{q}_{T-t}(X_t, V_t) \|^2 ] \\
+ L^2 \, \mathbb{E}_{\boldsymbol{Q}_T^{\leftarrow}} [ \| (X_{kh}, V_{kh}) - (X_t, V_t) \|^2 ] .
\end{aligned}
$$

By applying the moment bounds in Lemma 18 together with Lemma 19 on the movement of the CLD process, we obtain

$$
\begin{aligned}
\mathbb{E}_{\boldsymbol{Q}_T^{\leftarrow}} [ \| \mathbf{s}_{T-kh}(X_{kh}, V_{kh}) - \nabla_v \ln \boldsymbol{q}_{T-t}(X_t, V_t) \|^2 ] \\
\lesssim \varepsilon_{\mathrm{score}}^2 + L^2 dh + L^2 h^2 \, (d + \mathfrak{m}_2^2) + L^3 dh^2 + L^2 \, (dh + \mathfrak{m}_2^2 h^2) \\
\lesssim \varepsilon_{\mathrm{score}}^2 + L^2 dh + L^2 \mathfrak{m}_2^2 h^2 .
\end{aligned}
$$

The proof is concluded via an approximation argument as in Section B.2. $\qquad \square$

**Remark.** We now pause to discuss why the discretization bound above does not improve upon the result for DDPM (Theorem 10). In the context of log-concave sampling, one instead considers the underdamped Langevin process

$$
\mathrm{d}X_t = V_t \, ,
$$
$$
\mathrm{d}V_t = -\nabla U(X_t) \, \mathrm{d}t - \gamma \, V_t \, \mathrm{d}t + \sqrt{2\gamma} \, \mathrm{d}B_t \, ,
$$

which is discretized to yield the algorithm

$$
\mathrm{d}X_t = V_t \, ,
$$
$$
\mathrm{d}V_t = -\nabla U(X_{kh}) \, \mathrm{d}t - \gamma \, V_t \, \mathrm{d}t + \sqrt{2\gamma} \, \mathrm{d}B_t \, ,
$$

for $t \in [kh, (k+1)h]$. Let $\boldsymbol{P}_T$ denote the path measure for the algorithm, and let $\boldsymbol{Q}_T$ denote the path measure for the continuous-time process. After applying Girsanov's theorem, we obtain

$$
\mathsf{KL}(\boldsymbol{Q}_T \,\|\, \boldsymbol{P}_T) \asymp \frac{1}{\gamma} \sum_{k=0}^{N-1} \mathbb{E}_{\boldsymbol{Q}_T} \int_{kh}^{(k+1)h} \| \nabla U(X_t) - \nabla U(X_{kh}) \|^2 \, \mathrm{d}t \, .
$$

In this expression, note that $\nabla U$ depends only on the position coordinate. Since the $X$ process is smoother (as we do not add Brownian motion directly to $X$), the error $\| \nabla U(X_t) - \nabla U(X_{kh}) \|^2$ is of size $O(dh^2)$, which allows us to take step size $h \lesssim 1/\sqrt{d}$. This explains why the use of the underdamped Langevin diffusion leads to improved dimension dependence for log-concave sampling.

In contrast, consider the reverse process, in which

$$
\mathsf{KL}(\boldsymbol{Q}_T^{\leftarrow} \,\|\, \boldsymbol{P}_T^{q_T}) = 2 \sum_{k=0}^{N-1} \mathbb{E}_{\boldsymbol{Q}_T^{\leftarrow}} \int_{kh}^{(k+1)h} \| \boldsymbol{s}_{T-kh}(X_{kh}, V_{kh}) - \nabla_v \ln \boldsymbol{q}_{T-t}(X_t, V_t) \|^2 \, \mathrm{d}t \, .
$$

Since discretization of the reverse process involves the score function, which depends on both $X$ and $V$, the error now involves controlling $\|V_t - V_{kh}\|^2$, which is of size $O(dh)$ (the process $V$ is not very smooth because it includes a Brownian motion component). Therefore, from the form of the reverse process, we may expect that SGMs based on the CLD do not improve upon the dimension dependence of DDPM.

In Section C.5, we use this observation in order to prove a rigorous lower bound against discretization of SGMs based on the CLD.

## C.3  Proof of Theorem 7

*Proof of Theorem 7.* By the data processing inequality,

$$\mathsf{TV}(\boldsymbol{p}_T, \boldsymbol{q}_0) \le \mathsf{TV}(\boldsymbol{P}_T, \boldsymbol{P}_T^{q_T}) + \mathsf{TV}(\boldsymbol{P}_T^{q_T}, \boldsymbol{Q}_T^{\leftarrow}) \le \mathsf{TV}(\boldsymbol{q}_T, \boldsymbol{\gamma}^{2d}) + \mathsf{TV}(\boldsymbol{P}_T^{q_T}, \boldsymbol{Q}_T^{\leftarrow}) \,.$$

In Ma et al. (2021), following the entropic hypocoercivity approach of Villani (2009), Ma et al. consider a Lyapunov functional L which is equivalent to the sum of the KL divergence and the Fisher information,

$$\mathsf{L}(\boldsymbol{\mu} \parallel \boldsymbol{\gamma}^{2d}) \asymp \mathsf{KL}(\boldsymbol{\mu} \parallel \boldsymbol{\gamma}^{2d}) + \mathsf{FI}(\boldsymbol{\mu} \parallel \boldsymbol{\gamma}^{2d}) \,,$$

which decays exponentially fast in time: there exists a universal constant $c > 0$ such that for all $t \ge 0$,

$$\mathsf{L}(\boldsymbol{q}_t \parallel \boldsymbol{\gamma}^{2d}) \le \exp(-ct) \, \mathsf{L}(\boldsymbol{q}_0 \parallel \boldsymbol{\gamma}^{2d}) \,.$$

Since $\boldsymbol{q}_0 = q \otimes \gamma^d$ and $\boldsymbol{\gamma}^{2d} = \gamma^d \otimes \gamma^d$, then $\mathsf{L}(\boldsymbol{q}_0 \parallel \boldsymbol{\gamma}^{2d}) \lesssim \mathsf{KL}(q \parallel \gamma^d) + \mathsf{FI}(q \parallel \gamma^d)$. By Pinsker's inequality and Theorem 16, we deduce that

$$\mathsf{TV}(\boldsymbol{p}_T, \boldsymbol{q}_0) \lesssim \sqrt{\mathsf{KL}(q \parallel \gamma^d) + \mathsf{FI}(q \parallel \gamma^d)} \exp(-cT) + (\varepsilon_{\mathrm{score}} + L\sqrt{dh} + L\mathfrak{m}_2 h) \sqrt{T} \,,$$

which completes the proof. $\qquad\qquad\square$

## C.4  Auxiliary lemmas

We start with a perturbation lemma for the score function.

**Lemma 17** (score perturbation lemma). *Let $0 < \zeta < 1$. Suppose that $\boldsymbol{M}_0, \boldsymbol{M}_1 \in \mathbb{R}^{2d \times 2d}$ are two matrices, where $\boldsymbol{M}_1$ is symmetric. Also, assume that $\|\boldsymbol{M}_0 - \boldsymbol{I}_{2d}\|_{\mathrm{op}} \le \zeta$, so that $\boldsymbol{M}_0$ is invertible. Let $\boldsymbol{q} = \exp(-\boldsymbol{H})$ be a probability density on $\mathbb{R}^{2d}$ such that $\nabla \boldsymbol{H}$ is L-Lipschitz with $L \le \frac{1}{4\|\boldsymbol{M}_1\|_{\mathrm{op}}}$. Then, it holds that*

$$\left\| \nabla \ln \frac{(\boldsymbol{M}_0)_{\#}\boldsymbol{q} * \mathsf{normal}(0, \boldsymbol{M}_1)}{\boldsymbol{q}}(\boldsymbol{\theta}) \right\| \lesssim L\sqrt{\|\boldsymbol{M}_1\|_{\mathrm{op}} d} + L\zeta \|\boldsymbol{\theta}\| + (\zeta + L\|\boldsymbol{M}_1\|_{\mathrm{op}}) \|\nabla \boldsymbol{H}(\boldsymbol{\theta})\| \,.$$

*Proof.* The proof follows along the lines of Lee et al. (2022a, Lemma C.12). First, we show that when $\boldsymbol{M}_0 = \boldsymbol{I}_{2d}$, if $L \le \frac{1}{2\|\boldsymbol{M}_1\|_{\mathrm{op}}}$ then

$$\left\| \nabla \ln \frac{\boldsymbol{q} * \mathsf{normal}(0, \boldsymbol{M}_1)}{\boldsymbol{q}}(\boldsymbol{\theta}) \right\| \lesssim L\sqrt{\|\boldsymbol{M}_1\|_{\mathrm{op}} d} + L\|\boldsymbol{M}_1\|_{\mathrm{op}} \|\nabla \boldsymbol{H}(\boldsymbol{\theta})\| \,. \tag{20}$$

Let $\mathcal{S}$ denote the subspace $\mathcal{S} := \mathrm{range}\, \boldsymbol{M}_1$. Then, since

$$\big(\boldsymbol{q} * \mathsf{normal}(0, \boldsymbol{M}_1)\big)(\boldsymbol{\theta}) = \int_{\boldsymbol{\theta}+\mathcal{S}} \exp\big(-\frac{1}{2} \langle \boldsymbol{\theta} - \boldsymbol{\theta}', \boldsymbol{M}_1^{-1} (\boldsymbol{\theta} - \boldsymbol{\theta}') \rangle\big) \, \boldsymbol{q}(\mathrm{d}\boldsymbol{\theta}') \,,$$

where $\boldsymbol{M}_1^{-1}$ is well-defined on $\mathcal{S}$, we have

$$\begin{aligned}
&\left\| \nabla \ln \frac{\boldsymbol{q} * \mathsf{normal}(0, \boldsymbol{M}_1)}{\boldsymbol{q}}(\boldsymbol{\theta}) \right\| \\
&= \left\| \frac{\int_{\boldsymbol{\theta}+\mathcal{S}} \nabla \boldsymbol{H}(\boldsymbol{\theta}') \exp(-\frac{1}{2} \langle \boldsymbol{\theta} - \boldsymbol{\theta}', \boldsymbol{M}_1^{-1} (\boldsymbol{\theta} - \boldsymbol{\theta}') \rangle) \, \boldsymbol{q}(\mathrm{d}\boldsymbol{\theta}')}{\int_{\boldsymbol{\theta}+\mathcal{S}} \exp(-\frac{1}{2} \langle \boldsymbol{\theta} - \boldsymbol{\theta}', \boldsymbol{M}_1^{-1} (\boldsymbol{\theta} - \boldsymbol{\theta}') \rangle) \, \boldsymbol{q}(\mathrm{d}\boldsymbol{\theta}')} - \nabla \boldsymbol{H}(\boldsymbol{\theta}) \right\| \\
&= \|\mathbb{E}_{\boldsymbol{q}_{\boldsymbol{\theta}}} \nabla \boldsymbol{H} - \nabla \boldsymbol{H}(\boldsymbol{\theta})\| \,.
\end{aligned}$$

Here, $\boldsymbol{q}_{\boldsymbol{\theta}}$ is the measure on $\boldsymbol{\theta} + \mathcal{S}$ such that

$$\boldsymbol{q}_{\boldsymbol{\theta}}(\mathrm{d}\boldsymbol{\theta}') \propto \exp\big(-\frac{1}{2} \langle \boldsymbol{\theta} - \boldsymbol{\theta}', \boldsymbol{M}_1^{-1} (\boldsymbol{\theta} - \boldsymbol{\theta}') \rangle\big) \, \boldsymbol{q}(\mathrm{d}\boldsymbol{\theta}') \,.$$

Note that since $L \le \frac{1}{2\|\boldsymbol{M}_1\|_{\mathrm{op}}}$, then if we write $\boldsymbol{q}_{\boldsymbol{\theta}}(\boldsymbol{\theta}') \propto \exp(-\boldsymbol{H}_{\boldsymbol{\theta}}(\boldsymbol{\theta}'))$, we have

$$\nabla^2 \boldsymbol{H}_{\boldsymbol{\theta}} \succeq \big(\frac{1}{\|\boldsymbol{M}_1\|_{\mathrm{op}}} - L\big) I_d \succeq \frac{1}{2\|\boldsymbol{M}_1\|_{\mathrm{op}}} I_d \qquad \text{on } \boldsymbol{\theta} + \mathcal{S} \,.$$

Let $\boldsymbol{\theta}_\star \in \arg\min \boldsymbol{H}_{\boldsymbol{\theta}}$ denote a mode. We bound

$$\|\mathbb{E}_{\boldsymbol{q}_{\boldsymbol{\theta}}} \nabla \boldsymbol{H} - \nabla \boldsymbol{H}(\boldsymbol{\theta})\| \le L \,\mathbb{E}_{\boldsymbol{\theta}' \sim \boldsymbol{q}_{\boldsymbol{\theta}}}\|\boldsymbol{\theta}' - \boldsymbol{\theta}\| \le L \,\mathbb{E}_{\boldsymbol{\theta}' \sim \boldsymbol{q}_{\boldsymbol{\theta}}}\|\boldsymbol{\theta}' - \boldsymbol{\theta}_\star\| + L \,\|\boldsymbol{\theta}_\star - \boldsymbol{\theta}\| \,.$$

For the first term, (Dalalyan et al., 2019, Proposition 2) yields

$$\mathbb{E}_{\boldsymbol{\theta}' \sim \boldsymbol{q}_{\boldsymbol{\theta}}}\|\boldsymbol{\theta}' - \boldsymbol{\theta}_\star\| \le \sqrt{2 \,\|\boldsymbol{M}_1\|_{\mathrm{op}} \, d} \,.$$

For the second term, since the mode satisfies $\nabla \boldsymbol{H}(\boldsymbol{\theta}_\star) + \boldsymbol{M}_1^{-1}\,(\boldsymbol{\theta}_\star - \boldsymbol{\theta}) = 0$, we have

$$\|\boldsymbol{\theta}_\star - \boldsymbol{\theta}\| \le \|\boldsymbol{M}_1\|_{\mathrm{op}} \,\|\nabla \boldsymbol{H}(\boldsymbol{\theta}_\star)\| \le L \,\|\boldsymbol{M}_1\|_{\mathrm{op}} \,\|\boldsymbol{\theta}_\star - \boldsymbol{\theta}\| + \|\boldsymbol{M}_1\|_{\mathrm{op}} \,\|\nabla \boldsymbol{H}(\boldsymbol{\theta})\|$$

which is rearranged to yield

$$\|\boldsymbol{\theta}_\star - \boldsymbol{\theta}\| \le 2 \,\|\boldsymbol{M}_1\|_{\mathrm{op}} \,\|\nabla \boldsymbol{H}(\boldsymbol{\theta})\| \,.$$

After combining the bounds, we obtain the claimed estimate (20).

Next, we consider the case of general $\boldsymbol{M}_0$. We have

$$\left\|\nabla \ln \frac{(\boldsymbol{M}_0)_\# \boldsymbol{q} * \mathsf{normal}(0, \boldsymbol{M}_1)}{\boldsymbol{q}}(\boldsymbol{\theta})\right\|$$
$$\le \left\|\nabla \ln \frac{(\boldsymbol{M}_0)_\# \boldsymbol{q} * \mathsf{normal}(0, \boldsymbol{M}_1)}{(\boldsymbol{M}_0)_\# \boldsymbol{q}}(\boldsymbol{\theta})\right\| + \left\|\nabla \ln \frac{(\boldsymbol{M}_0)_\# \boldsymbol{q}}{\boldsymbol{q}}(\boldsymbol{\theta})\right\| \,.$$

We can apply (20) with $(\boldsymbol{M}_0)_\# \boldsymbol{q}$ in place of $\boldsymbol{q}$, noting that $(\boldsymbol{M}_0)_\# \boldsymbol{q} \propto \exp(-\boldsymbol{H}')$ for $\boldsymbol{H}' \coloneqq \boldsymbol{H} \circ \boldsymbol{M}_0$ which is $L'$-smooth for $L' \coloneqq L \,\|\boldsymbol{M}_0\|_{\mathrm{op}}^2 \lesssim L$, to get

$$\left\|\nabla \ln \frac{(\boldsymbol{M}_0)_\# \boldsymbol{q} * \mathsf{normal}(0, \boldsymbol{M}_1)}{(\boldsymbol{M}_0)_\# \boldsymbol{q}}(\boldsymbol{\theta})\right\| \lesssim L\sqrt{\|\boldsymbol{M}_1\|_{\mathrm{op}} \, d} + L \,\|\boldsymbol{M}_1\|_{\mathrm{op}} \,\|\boldsymbol{M}_0 \nabla \boldsymbol{H}(\boldsymbol{M}_0 \boldsymbol{\theta})\|$$
$$\lesssim L\sqrt{\|\boldsymbol{M}_1\|_{\mathrm{op}} \, d} + L \,\|\boldsymbol{M}_1\|_{\mathrm{op}} \,\|\nabla \boldsymbol{H}(\boldsymbol{M}_0 \boldsymbol{\theta})\| \,.$$

Note that

$$\|\nabla \boldsymbol{H}(\boldsymbol{M}_0 \boldsymbol{\theta})\| \le \|\nabla \boldsymbol{H}(\boldsymbol{\theta})\| + L \,\|(\boldsymbol{M}_0 - \boldsymbol{I}_{2d})\,\boldsymbol{\theta}\| \lesssim \|\nabla \boldsymbol{H}(\boldsymbol{\theta})\| + L\zeta \,\|\boldsymbol{\theta}\| \,.$$

We also have

$$\left\|\nabla \ln \frac{(\boldsymbol{M}_0)_\# \boldsymbol{q}}{\boldsymbol{q}}(\boldsymbol{\theta})\right\| = \|\boldsymbol{M}_0 \nabla \boldsymbol{H}(\boldsymbol{M}_0 \boldsymbol{\theta}) - \nabla \boldsymbol{H}(\boldsymbol{\theta})\|$$
$$\le \|\boldsymbol{M}_0 \nabla \boldsymbol{H}(\boldsymbol{M}_0 \boldsymbol{\theta}) - \boldsymbol{M}_0 \nabla \boldsymbol{H}(\boldsymbol{\theta})\| + \|\boldsymbol{M}_0 \nabla \boldsymbol{H}(\boldsymbol{\theta}) - \nabla \boldsymbol{H}(\boldsymbol{\theta})\|$$
$$\lesssim L \,\|(\boldsymbol{M}_0 - \boldsymbol{I}_{2d})\,\boldsymbol{\theta}\| + \zeta \,\|\nabla \boldsymbol{H}(\boldsymbol{\theta})\| \lesssim L\zeta \,\|\boldsymbol{\theta}\| + \zeta \,\|\nabla \boldsymbol{H}(\boldsymbol{\theta})\| \,.$$

Combining the bounds,

$$\left\|\nabla \ln \frac{(\boldsymbol{M}_0)_\# \boldsymbol{q} * \mathsf{normal}(0, \boldsymbol{M}_1)}{\boldsymbol{q}}(\boldsymbol{\theta})\right\|$$
$$\lesssim L\sqrt{\|\boldsymbol{M}_1\|_{\mathrm{op}} \, d} + L\zeta \,(1 + L \,\|\boldsymbol{M}_1\|_{\mathrm{op}}) \,\|\boldsymbol{\theta}\| + (\zeta + L \,\|\boldsymbol{M}_1\|_{\mathrm{op}}) \,\|\nabla \boldsymbol{H}(\boldsymbol{\theta})\|$$
$$\lesssim L\sqrt{\|\boldsymbol{M}_1\|_{\mathrm{op}} \, d} + L\zeta \,\|\boldsymbol{\theta}\| + (\zeta + L \,\|\boldsymbol{M}_1\|_{\mathrm{op}}) \,\|\nabla \boldsymbol{H}(\boldsymbol{\theta})\|$$

so the lemma follows. $\qquad\square$

Next, we prove the moment and movement bounds for the CLD.

**Lemma 18** (moment bounds for CLD). *Suppose that Assumptions 2 and 4 hold. Let $(\bar{X}_t, \bar{V}_t)_{t \in [0,T]}$ denote the forward process* (6)*.*

1. *(moment bound) For all $t \ge 0$,*

$$\mathbb{E}[\|(\bar{X}_t, \bar{V}_t)\|^2] \lesssim d + \mathfrak{m}_2^2 \,.$$

2. *(score function bound) For all $t \geq 0$,*

$$\mathbb{E}[\|\nabla \ln \boldsymbol{q}_t(\bar{X}_t, \bar{V}_t)\|^2] \leq Ld.$$

*Proof.*     1. We can write

$$\mathbb{E}[\|(\bar{X}_t, \bar{V}_t)\|^2] = W_2^2(\boldsymbol{q}_t, \delta_{\mathbf{0}}) \lesssim W_2^2(\boldsymbol{q}_t, \boldsymbol{\gamma}^{2d}) + W_2^2(\boldsymbol{\gamma}^{2d}, \delta_{\mathbf{0}}) \lesssim d + W_2^2(\boldsymbol{q}_t, \boldsymbol{\gamma}^{2d}).$$

Next, the coupling argument of Cheng et al. (2018) shows that the CLD converges exponentially fast in the Wasserstein metric associated to a twisted norm $\|\|\cdot\|\|$ which is equivalent (up to universal constants) to the Euclidean norm $\|\cdot\|$. It implies the following result, see, e.g., Cheng et al. (2018, Lemma 8):

$$W_2^2(\boldsymbol{q}_t, \boldsymbol{\gamma}^{2d}) \lesssim W_2^2(\boldsymbol{q}, \boldsymbol{\gamma}^{2d}) \lesssim W_2^2(\boldsymbol{q}, \delta_{\mathbf{0}}) + W_2^2(\delta_{\mathbf{0}}, \boldsymbol{\gamma}^{2d}) \lesssim d + \mathfrak{m}_2^2.$$

2. The proof is the same as in Lemma 11.

$\square$

**Lemma 19** (movement bound for CLD). *Suppose that Assumptions 2 holds. Let $(\bar{X}_t, \bar{V}_t)_{t \in [0,T]}$ denote the forward process* (6). *For $0 < s < t$ with $\delta := t - s$, if $\delta \leq 1$,*

$$\mathbb{E}[\|(\bar{X}_t, \bar{V}_t) - (\bar{X}_s, \bar{V}_s)\|^2] \lesssim \delta^2 \mathfrak{m}_2^2 + \delta d.$$

*Proof.* First,

$$\mathbb{E}[\|\bar{X}_t - \bar{X}_s\|^2] = \mathbb{E}\Big[\Big\|\int_s^t \bar{V}_r \, \mathrm{d}r\Big\|^2\Big] \leq \delta \int_s^t \mathbb{E}[\|\bar{V}_r\|^2] \, \mathrm{d}r \lesssim \delta^2 \, (d + \mathfrak{m}_2^2),$$

where we used the moment bound in Lemma 18. Next,

$$\mathbb{E}[\|\bar{V}_t - \bar{V}_s\|^2] = \mathbb{E}\Big[\Big\|\int_s^t (-\bar{X}_r - 2\bar{V}_r) \, \mathrm{d}r + 2\,(B_t - B_s)\Big\|^2\Big] \lesssim \delta \int_s^t \mathbb{E}[\|\bar{X}_r\|^2 + \|\bar{V}_r\|^2] \, \mathrm{d}r + \delta d$$

$$\lesssim \delta^2 \, (d + \mathfrak{m}_2^2) + \delta d,$$

where we used Lemma 18 again.

$\square$

### C.5   Lower bound against CLD

When proving upper bounds on the KL divergence, we can use the approximation argument described in Section B.2 in order to invoke Girsanov's theorem. However, when proving lower bounds on the KL divergence, this approach no longer works, so we check Novikov's condition directly for the setting of Theorem 8.

**Lemma 20** (Novikov's condition holds for CLD). *Consider the setting of Theorem 8. Then, Novikov's condition 15 holds.*

We defer the proof of Lemma 20 to the end of this section. Admitting Lemma 20, we now prove Theorem 8.

*Proof of Theorem 8.* Since $\boldsymbol{q}_0 = \gamma^d \otimes \gamma^d = \boldsymbol{\gamma}^{2d}$ is stationary for the forward process (6), we have $\boldsymbol{q}_t = \boldsymbol{\gamma}^{2d}$ for all $t \geq 0$. In this proof, since the score estimate is perfect and $\boldsymbol{q}_T = \boldsymbol{\gamma}^{2d}$, we simply denote the path measure for the algorithm as $\boldsymbol{P}_T = \boldsymbol{P}_T^{q_T}$. From Girsanov's theorem in the form of Corollary 15 and from $\boldsymbol{s}_{T-kh}(x,v) = \nabla_v \ln \boldsymbol{q}_{T-kh}(x,v) = -v$, we have

$$\mathsf{KL}(\boldsymbol{Q}_T^{\leftarrow} \,\|\, \boldsymbol{P}_T) = 2 \sum_{k=0}^{N-1} \mathbb{E}_{\boldsymbol{Q}_T^{\leftarrow}} \int_{kh}^{(k+1)h} \|V_{kh} - V_t\|^2 \, \mathrm{d}t. \tag{21}$$

To lower bound this quantity, we use the inequality $\|x + y\|^2 \geq \frac{1}{2} \|x\|^2 - \|y\|^2$ to write, for $t \in [kh, (k+1)h]$

$$\mathbb{E}_{\boldsymbol{Q}_T^\leftarrow}[\|V_{kh} - V_t\|^2] = \mathbb{E}[\|\bar{V}_{T-kh} - \bar{V}_{T-t}\|^2]$$

$$= \mathbb{E}\left[\left\|\int_{T-t}^{T-kh} \{-\bar{X}_s - 2\bar{V}_s\} \, \mathrm{d}s + 2\left(B_{T-kh} - B_{T-t}\right)\right\|^2\right]$$

$$\geq 2\,\mathbb{E}[\|B_{T-kh} - B_{T-t}\|^2] - \mathbb{E}\left[\left\|\int_{T-t}^{T-kh} \{-\bar{X}_s - 2\bar{V}_s\} \, \mathrm{d}s\right\|^2\right]$$

$$\geq 2d\,(t - kh) - (t - kh) \int_{T-t}^{T-kh} \mathbb{E}[\|\bar{X}_s + 2\bar{V}_s\|^2] \, \mathrm{d}s$$

$$\geq 2d\,(t - kh) - (t - kh) \int_{T-t}^{T-kh} \mathbb{E}[2\|\bar{X}_s\|^2 + 8\|\bar{V}_s\|^2] \, \mathrm{d}s\,.$$

Using the fact that $\bar{X}_s \sim \gamma^d$ and $\bar{V}_s \sim \gamma^d$ for all $s \in [0, T]$, we can then bound

$$\mathbb{E}_{\boldsymbol{Q}_T^\leftarrow}[\|V_{kh} - V_t\|^2] \geq 2d\,(t - kh) - 10d\,(t - kh)^2 \geq d\,(t - kh)\,,$$

provided that $h \leq \frac{1}{10}$. Substituting this into (21),

$$\mathsf{KL}(\boldsymbol{Q}_T^\leftarrow \,\|\, \boldsymbol{P}_T) \geq 2d \sum_{k=0}^{N-1} \int_{kh}^{(k+1)h} (t - kh)^2 \, \mathrm{d}t = dh^2 N = dhT\,.$$

This proves the result. $\qquad\square$

This lower bound shows that the Girsanov discretization argument of Theorem 16 is essentially tight (except possibly the dependence on $L$).

We now prove Lemma 20.

*Proof of Lemma 20.* Similarly to the proof of Theorem 8 above, we note that

$$\|\boldsymbol{s}_{T-kh}(X_{kh}, V_{kh}) - \nabla_v \ln \boldsymbol{q}_{T-t}(X_t, V_t)\|^2 = \|\bar{V}_{T-kh} - \bar{V}_{T-t}\|^2$$

$$= \left\|\int_{T-t}^{T-kh} \{-\bar{X}_s - 2\bar{V}_s\} \, \mathrm{d}s + 2\left(B_{T-kh} - B_{T-t}\right)\right\|^2$$

$$\lesssim h^2 \sup_{s \in [0,T]} (\|\bar{X}_s\|^2 + \|\bar{V}_s\|^2) + \sup_{s \in [T-(k+1)h, T-kh]} \|B_{T-kh} - B_s\|^2\,.$$

Hence, for a universal constant $C > 0$ (which may change from line to line)

$$\mathbb{E}_{\boldsymbol{Q}_T^\leftarrow} \exp\left(2 \sum_{k=0}^{N-1} \int_{kh}^{(k+1)h} \|\boldsymbol{s}_{T-kh}(X_{kh}, V_{kh}) - \nabla_v \ln \boldsymbol{q}_{T-t}(X_t, V_t)\|^2 \, \mathrm{d}t\right)$$

$$\leq \mathbb{E} \exp\left(CTh^2 \sup_{s \in [0,T]} (\|\bar{X}_s\|^2 + \|\bar{V}_s\|^2) + Ch \sum_{k=0}^{N-1} \sup_{s \in [T-(k+1)h, T-kh]} \|B_{T-kh} - B_s\|^2\right)\,.$$

By the Cauchy–Schwarz inequality, to prove that this expectation is finite, it suffices to consider the two terms in the exponential separately.

Next, we recall that

$$\mathrm{d}\bar{X}_t = \bar{V}_t \, \mathrm{d}t\,,$$

$$\mathrm{d}\bar{V}_t = -(\bar{X}_t + 2\bar{V}_t) \, \mathrm{d}t + 2 \, \mathrm{d}B_t\,.$$

Define $\bar{Y}_t := \bar{X}_t + \bar{V}_t$. Then, $\mathrm{d}\bar{Y}_t = -\bar{Y}_t \, \mathrm{d}t + 2 \, \mathrm{d}B_t$, which admits the explicit solution

$$\bar{Y}_t = \exp(-t)\,\bar{Y}_0 + 2 \int_0^t \exp\{-(t - s)\} \, \mathrm{d}B_s\,.$$

Also, $d\bar{X}_t = -\bar{X}_t\,dt + \bar{Y}_t\,dt$, which admits the solution

$$\bar{X}_t = \exp(-t)\,\bar{X}_0 + \int_0^t \exp\{-(t-s)\}\,\bar{Y}_t\,dt\,.$$

Hence,

$$\|\bar{X}_t\| + \|\bar{V}_t\| \le 2\,\|\bar{X}_t\| + \|\bar{Y}_t\| \lesssim \|\bar{X}_0\| + \sup_{s\in[0,T]}\|\bar{Y}_s\|$$

and

$$\sup_{t\in[0,T]}\|\bar{Y}_t\| \lesssim \|\bar{X}_0\| + \|\bar{V}_0\| + \sup_{t\in[0,T]}\left\{\exp(-t)\left\|\int_0^t \exp(s)\,dB_s\right\|\right\}$$

$$= \|\bar{X}_0\| + \|\bar{V}_0\| + \sup_{t\in[0,T]}\exp(-t)\,\|\tilde{B}_{(\exp(2t)-1)/2}\|$$

where $\tilde{B}$ is another standard Brownian motion and we use the interpretation of stochastic integrals as time changes of Brownian motion (Steele, 2001, Corollary 7.1). Since $(\bar{X}_0, \bar{V}_0) \sim \gamma^{2d}$ has independent entries, then

$$\mathbb{E}\exp(CTh^2\,\{\|\bar{X}_0\|^2 + \|\bar{V}_0\|^2\}) = \prod_{j=1}^d \mathbb{E}\exp(CTh^2\,\langle e_j, \bar{X}_0\rangle^2)\,\mathbb{E}\exp(CTh^2\,\langle e_j, \bar{V}_0\rangle^2) < \infty$$

provided that $h \lesssim 1/\sqrt{T}$. Also, by the Cauchy–Schwarz inequality, we can give a crude bound: writing $\tau(t) = (\exp(2t)-1)/2$,

$$\mathbb{E}\exp\left(CTh^2\sup_{t\in[0,T]}\exp(-2t)\,\|\tilde{B}_{\tau(t)}\|^2\right)$$

$$\le \left[\mathbb{E}\exp\left(2CTh^2\sup_{t\in[0,1]}\exp(-2t)\,\|\tilde{B}_{\tau(t)}\|^2\right)\right]^{1/2}$$

$$\times \left[\mathbb{E}\exp\left(2CTh^2\sup_{t\in[1,T]}\exp(-2t)\,\|\tilde{B}_{\tau(t)}\|^2\right)\right]^{1/2}$$

where, by standard estimates on the supremum of Brownian motion (see, e.g., Chewi et al., 2021b, Lemma 23), the first factor is finite if $h \lesssim 1/\sqrt{T}$ (again using independence across the dimensions). For the second factor, if we split the sum according to $\exp(-2t) \asymp 2^k$ and use Hölder's inequality,

$$\mathbb{E}\exp\left(CTh^2\sup_{t\in[1,T]}\exp(-2t)\,\|\tilde{B}_{\tau(t)}\|^2\right)$$

$$\le \prod_{k=1}^K \left[\mathbb{E}\exp\left(CKTh^2\sup_{2^k\le t\le 2^{k+1}}\exp(-2t)\,\|\tilde{B}_{\tau(t)}\|^2\right)\right]^{1/K}$$

where $K = O(T)$. Then,

$$\mathbb{E}\exp\left(CT^2h^2\sup_{2^k\le t\le 2^{k+1}}\exp(-2t)\,\|\tilde{B}_{\tau(t)}\|^2\right)$$

$$\le \mathbb{E}\exp\left(CT^2h^2 2^{-k}\sup_{1\le t\le 2^{k+1}}\|\tilde{B}_{\tau(t)}\|^2\right) < \infty\,,$$

provided $h \lesssim 1/T$, where we again use Chewi et al. (2021b, Lemma 23) and split across the coordinates. The Cauchy–Schwarz inequality then implies

$$\mathbb{E}\exp\left(CTh^2\sup_{s\in[0,T]}(\|\bar{X}_s\|^2 + \|\bar{V}_s\|^2)\right) < \infty\,.$$

For the second term, by independence of the increments,

$$\mathbb{E}\exp\left(Ch\sum_{k=0}^{N-1}\sup_{s\in[T-(k+1)h,\,T-kh]}\|B_{T-kh} - B_s\|^2\right)$$

$$= \prod_{k=0}^{N-1}\mathbb{E}\exp\left(Ch\sup_{s\in[T-(k+1)h,\,T-kh]}\|B_{T-kh} - B_s\|^2\right) = \left[\mathbb{E}\exp\left(Ch\sup_{s\in[0,h]}\|B_s\|^2\right)\right]^N\,.$$

By Chewi et al. (2021b, Lemma 23), this quantity is finite if $h \lesssim 1$, which completes the proof. $\quad\square$

## D DERIVATION OF THE SCORE MATCHING OBJECTIVE

In this section, we present a self-contained derivation of the score matching objective (10) for the reader's convenience. See also Hyvärinen (2005); Vincent (2011); Song & Ermon (2019).

Recall that the problem is to solve

$$\underset{s_t \in \mathscr{F}}{\text{minimize}} \quad \mathbb{E}_{q_t}[\|s_t - \nabla \ln q_t\|^2].$$

This objective cannot be evaluated, even if we replace the expectation over $q_t$ with an empirical average over samples from $q_t$. The trick is to use an integration by parts identity to reformulate the objective. Here, $C$ will denote any constant that does not depend on the optimization variable $s_t$. Expanding the square,

$$\mathbb{E}_{q_t}[\|s_t - \nabla \ln q_t\|^2] = \mathbb{E}_{q_t}[\|s_t\|^2 - 2\langle s_t, \nabla \ln q_t\rangle] + C.$$

We can rewrite the second term using integration by parts:

$$\int \langle s_t, \nabla \ln q_t \rangle \, \mathrm{d}q_t = \int \langle s_t, \nabla q_t \rangle = -\int (\operatorname{div} s_t) \, \mathrm{d}q_t$$

$$= -\iint (\operatorname{div} s_t)\big(\exp(-t)\, x_0 + \sqrt{1 - \exp(-2t)}\, z_t\big)\, \mathrm{d}q(x_0)\, \mathrm{d}\gamma^d(z_t),$$

where $\gamma^d = \mathsf{normal}(0, I_d)$ and we used the explicit form of the law of the OU process at time $t$. Recall the Gaussian integration by parts identity: for any vector field $v : \mathbb{R}^d \to \mathbb{R}^d$,

$$\int (\operatorname{div} v)\, \mathrm{d}\gamma^d = \int \langle x, v(x)\rangle \, \mathrm{d}\gamma^d(x).$$

Applying this identity,

$$\int \langle s_t, \nabla \ln q_t\rangle \, \mathrm{d}q_t = -\frac{1}{\sqrt{1 - \exp(-2t)}} \int \langle z_t, s_t(x_t)\rangle \, \mathrm{d}q(x_0)\, \mathrm{d}\gamma^d(z_t)$$

where $x_t = \exp(-t)\, x_0 + \sqrt{1 - \exp(-2t)}\, z_t$. Substituting this in,

$$\mathbb{E}_{q_t}[\|s_t - \nabla \ln q_t\|^2] = \mathbb{E}\Big[\|s_t(X_t)\|^2 + \frac{2}{\sqrt{1 - \exp(-2t)}} \langle Z_t, s_t(X_t)\rangle\Big] + C$$

$$= \mathbb{E}\Big[\Big\|s(X_t) + \frac{1}{\sqrt{1 - \exp(-2t)}} Z_t\Big\|^2\Big] + C,$$

where $X_0 \sim q$ and $Z_t \sim \gamma^d$ are independent, and $X_t := \exp(-t)\, X_0 + \sqrt{1 - \exp(-2t)}\, Z_t$.

## E DEFERRED PROOFS

**Lemma 21.** *Suppose that* $\operatorname{supp} q \subseteq \mathsf{B}(0, R)$ *where* $R \geq 1$*, and let* $q_t$ *denote the law of the OU process at time* $t$*, started at* $q$*. Let* $\varepsilon > 0$ *be such that* $\varepsilon \ll \sqrt{d}$ *and set* $t \asymp \varepsilon^2/(\sqrt{d}\,(R \vee \sqrt{d}))$*. Then,*

1. $W_2(q_t, q) \leq \varepsilon$.

2. $q_t$ *satisfies*

$$\mathsf{KL}(q_t \parallel \gamma^d) \lesssim \frac{\sqrt{d}\,(R \vee \sqrt{d})^3}{\varepsilon^2}.$$

3. *For every* $t' \geq t$*,* $q_{t'}$ *satisfies Assumption 1 with*

$$L \lesssim \frac{dR^2\,(R \vee \sqrt{d})^2}{\varepsilon^4}.$$

*Proof.*  1. For the OU process (1), we have $\bar{X}_t := \exp(-t)\,\bar{X}_0 + \sqrt{1 - \exp(-2t)}\,Z$, where $Z \sim \mathsf{normal}(0, I_d)$ is independent of $\bar{X}_0$. Hence, for $t \lesssim 1$,

$$W_2^2(q, q_t) \leq \mathbb{E}\big[\big\|\big(1 - \exp(-t)\big)\bar{X}_0 + \sqrt{1 - \exp(-2t)}\,Z\big\|^2\big]$$
$$= \big(1 - \exp(-t)\big)^2 \mathbb{E}[\|\bar{X}_0\|^2] + \big(1 - \exp(-2t)\big)\,d \lesssim R^2 t^2 + dt\,.$$

We now take $t \lesssim \min\{\varepsilon/R, \varepsilon^2/d\}$ to ensure that $W_2^2(q, q_t) \leq \varepsilon^2$. Since $\varepsilon \ll \sqrt{d}$, it suffices to take $t \asymp \varepsilon^2/(\sqrt{d}\,(R \vee \sqrt{d}))$.

2. For this, we use the short-time regularization result in Otto & Villani (2001, Corollary 2), which implies that

$$\mathsf{KL}(q_t \,\|\, \gamma^d) \leq \frac{W_2^2(q, \gamma^d)}{4t} \lesssim \frac{W_2^2(q, \delta_0) + W_2^2(\gamma^d, \delta_0)}{t} \lesssim \frac{\sqrt{d}\,(R \vee \sqrt{d})^3}{\varepsilon^2}\,.$$

3. Using Mikulincer & Shenfeld (2022, Lemma 4), along the OU process,

$$\frac{1}{1 - \exp(-2t)}\,I_d - \frac{\exp(-2t)\,R^2}{(1 - \exp(-2t))^2}\,I_d \preccurlyeq -\nabla^2 \ln q_t(x) \preccurlyeq \frac{1}{1 - \exp(-2t)}\,I_d\,.$$

With our choice of $t$, it implies

$$\|\nabla^2 \ln q_{t'}\|_{\mathrm{op}} \lesssim \frac{1}{1 - \exp(-2t')} \vee \frac{\exp(-2t')\,R^2}{(1 - \exp(-2t'))^2} \lesssim \frac{1}{t} \vee \frac{R^2}{t^2} \lesssim \frac{dR^2\,(R \vee \sqrt{d})^2}{\varepsilon^4}\,.$$

$\square$

