# OpenReview forum: "Sampling is as easy as learning the score: theory for diffusion models with minimal data assumptions"
_ICLR.cc/2023/Conference — ICLR 2023 notable top 5%_

### Official Review · Reviewer_71Gb · 2022-10-21

**Confidence:** 4
**Correctness:** 3
**Technical Novelty And Significance:** 3
**Empirical Novelty And Significance:** Not applicable
**Recommendation:** 8

**Clarity, Quality, Novelty And Reproducibility:**

The presentation is clear except for a few minor issues below:

- The statement "LSI is slightly weaker than log-concavity" is misleading.

- Footnote on p.3: The error is fixed in the lastest version of (Block et al. 2020); see the new Proposition 21.

- I'm not sure why $\sigma_t$ is needed in Section 2.1. The authors never invoked $\sigma_t\neq \sqrt{2}$.

- The result of the overdamped Langevin process on p.5 is only for strongly log-concave distributions. The general implication is unclear.


**Strength And Weaknesses:**

Strength:

- The presentation is clear.

- The proof ideas are simple and fairly straightforward. As the authors mentioned, they seem robust enough to generalize to other settings beyond the OU process considered in the present work.

Weaknesses:

- This paper does not really deal with DDPM in the sense of the cited (Song et al. 2021b). First, at least in (Song et al. 2021b), DDPM refers to a specific SDE with time-inhomogeneous diffusion. Although the authors briefly mention that the analysis can be extended to that case, I don't think it is such a straightforward matter. For instance,  the first step in Section 4 would break down, as $\tilde{p}_0 \neq \gamma^d$. How do we handle this?

    Another major difference is, in (Song et al. 2021b), the sampling part is done by a predictor-corrector sampler; see Appendix G. Incorporating this source of error is quite important as it does reduce the variance of the sampling procedure significantly. Instead, the authors opted for an exact sampler by taking advantage of the time-inhomogeneous OU process, which, echoing the above, is not available for DDPM.



- One thing that is unclear to me is the scaling issue as follows. Since both TV and KL are invariant to scaling, instead of $q(x)$ one may consider a new measure $q(cx)$ for any $c >0$. Inspecting the bound in Theorem 2, it is clear that the first term is invariant, the second moment in the second term scales accordingly, whereas the $\varepsilon_{\text{score}}$ term is again left as constant. On the other hand, intuitively, as $c\rightarrow \infty$ the error in the score estimate might shrink (since this corresponds to shrinking the moments of $q$).

    To conclude, the bound in Theorem 2 doesn't seem to capture the scale invariance of TV/KL, suggesting that there might be some artifact in the proof. (That being said, I acknowledge that my argument above is no less vague than the authors', so this should not be taken as a major criticism.)

- I find the argument on the deficiency of underdamped Langevin fairly weak. On one hand, as the authors have noticed, no definitive statement is given. On the other hand, all of the bounds in this paper are either given by KL or TV, but these two are not really the "right" metric for underdamped Langevin (Desvillettes and Villani 2000).

- The comparison to (De Bortoli 2022) is not entirely fair since the convergence metric there is given in the Wasserstein distance, which makes sense under the manifold hypothesis. The Wasserstein distance is ideal for this setting as it does not rely on the measures being absolutely continuous to each other.

- As the authors acknowledged, a serious limitation is that the score estimate part is assumed away, whereas in practice estimating the score is the bottleneck.



Desvillettes and Villani 2000, On the trend to global equilibrium in spatially inhomogeneous entropy-dissipating systems: The linear Fokker-Planck equation.

**Summary Of The Paper:**

This paper examines the convergence of SGMs under fairly general assumptions. The main novelty compared to existing works is the $L^2$ error assumption on the score estimates, as opposed to the more common $L^\infty$.The main conclusion is that one can basically reduce the difficulty of SGMs to estimating the scores since the sampling part afterward is relatively easy: the authors proved a convergence bound that scales polynomially with respect to the relevant parameters. Extention to CLD is discussed.

**Summary Of The Review:**

This paper provides several convergence results for diffusion models with OU process. I think this is a solid paper, but can be significantly improved if the authors can:

1. prove results on DDPM or other SDEs in (Song et al. 2021b).

2. provide more context on the related work, for instance (De Bortoli 2022).

3. figure out the scaling issue.

4. improve the section on CLD.

---

> ### Author Response · Authors · 2022-11-15
> **Our answers, Part 1/2**
>
> Thank you for your review. Below we address some of your points.
>
> Overall, although we agree with your clarifying remarks, we believe that **many of your criticisms are unfair as they suggest that we should resolve all theoretical issues regarding SGMs,** which is out of scope of our submission. As our work is **the first to provide polynomial guarantees for SGMs under minimal data assumptions**, *we believe that this already constitutes a substantial advance in this area and we hope that you will reconsider your score if you believe that we have addressed your concerns fairly.*
>
> *This paper does not really deal with DDPM in the sense of the cited (Song et al. 2021b).*
>
> Thank you for pointing this out. You are correct that we do not consider precisely the setting in Song et al., but we do not believe this poses serious difficulties. For example, as mentioned in the Song et al. paper underneath eq. (11), DDPM corresponds to a discretization of a reverse SDE which has the property that even for different choices of the time-inhomogeneous diffusion, the limiting distribution is still a standard Gaussian, so **we can still take $\tilde p_0 = \gamma^d$,** the standard multivariate Gaussian distribution.
>
> In our paper, we have chosen to study the *simplest model of generative modeling via reverse SDEs*. Although we could have incorporated different scalings of the noise and predictor-corrector steps (at the expense of longer proofs; as you point out, our proof techniques are fairly robust to these changes), and these choices do affect the precise quantitative dependencies of the result, they do not affect the core takeaway of our paper, which is that we can obtain polynomial-time guarantees for SGMs assuming accurate score estimation. **Our goal is not to provide a comprehensive analysis of all SGMs available in the literature but rather to convey our core takeaway and to lay the foundation for future theoretical studies of SGMs, and hence we believe that your suggestions, while relevant, are ultimately out of scope.**
>
> In our revision we will clearly point out the differences in our setting with that of Song et al. and emphasize that we make simplifying assumptions for clarity.
>
> *One thing that is unclear to me is the scaling issue as follows.*
>
> Thank you for raising this question. However, **it is not clear to us that the first term should be invariant**, since the KL divergence to the standard Gaussian will change if we replace $q$ by a scaling of $q$. In order to leave the KL divergence unchanged, we should also change the standard Gaussian to a Gaussian of different variance, corresponding to a different scaling of the forward/reverse processes, and this would then affect all of our quantitative bounds.
>
> For example, the first term would involve a different exponential rate $\exp(-c_0 t)$. The **second term would also be multiplied by another constant $c_1$ from the statement of Girsanov’s theorem**, as well as the change in $L, m_2$ (Lipschitz constant and second moment). Furthermore, the interpretation of $\varepsilon_{\rm score}$ would also change due to the **change in difficulty in estimating the score function** (even if the score were estimated with a simple law of large numbers, then the accuracy of the estimation would depend on the variance of the samples).
>
> *On the other hand, all of the bounds in this paper are either given by KL or TV, but these two are not really the "right" metric for underdamped Langevin (Desvillettes and Villani 2000).*
>
> Thank you for the comment. This is an interesting point. However, the reference that you sent indeed studies the underdamped Langevin diffusion using the KL divergence. Moreover, there is a successful history of KL divergence-based proofs in this context, see the monograph on hypocoercivity (Villani 2009), as well as the sampling reference Ma et al. (2021). Could you please elaborate on why KL is not the right metric here?
>
> *The comparison to (De Bortoli 2022) is not entirely fair since the convergence metric there is given in the Wasserstein distance, which makes sense under the manifold hypothesis. The Wasserstein distance is ideal for this setting as it does not rely on the measures being absolutely continuous to each other.*
>
> Thank you for bringing this up. However, we note that our guarantee in Corollary 4 (for compactly supported data distributions) can be translated into guarantees in the Wasserstein metric simply by using transport inequalities, e.g., the Lemma 9 in the reference below.
>
> https://infoscience.epfl.ch/record/297326

---

> > ### Author Response · Authors · 2022-11-15
> > **Our answers, Part 2/2**
> >
> > *As the authors acknowledged, a serious limitation is that the score estimate part is assumed away, whereas in practice estimating the score is the bottleneck.*
> >
> > Although this is indeed a limitation of our work, some assumption on the success of the score estimation procedure is obviously necessary to obtain sampling guarantees for SGMs at the level of generality of our results. Note that our goal is **not to provide a specific example** of a situation in which SGMs succeed, **but to provide general conditions** which imply their good performance. Although the reason why score estimation succeeds may be different in different contexts, our guarantees are generic in that they apply equally well regardless.
> >
> > We agree that score estimation is the bottleneck in practice, but this does not mean the sampling part is trivial. We emphasize that before our work, even with the assumption of accurate score estimation, **no polynomial time guarantees were known for the sampling portion of SGMs, and our main goal was to close this gap.**
> >
> >
> > **We guess that the following comments are MINOR, so we provide short answers; let us know if you are satisfied.**
> >
> > *The statement "LSI is slightly weaker than log-concavity" is misleading.*
> >
> > To be clear, LSI is slightly weaker than strong log-concavity; we will clarify this in the text.
> >
> > *Footnote on p.3: The error is fixed in the latest version of (Block et al. 2020); see the new Proposition 21.*
> >
> > We also noticed that (Block et al. 2020) has been updated and fixed: we will remove the footnote accordingly.
> >
> > *I'm not sure why $\sigma_t$ is needed in Section 2.1.*
> >
> > Actually, $\sigma_t$ is a degenerate matrix for the CLD case, and we need the general formula for SDE time reversal to justify the reversed SDE for CLD.
> >
> > *The result of the overdamped Langevin process on p.5 is only for strongly log-concave distributions. The general implication is unclear.*
> >
> > We will clarify this paragraph to emphasize that this remark holds only for strongly log-concave distributions.

---

> > > ### Comment · Reviewer_71Gb · 2022-11-27
> > > **Thank you for the detailed answers.**
> > >
> > > I thank the authors for the detailed rebuttal. Several concerns were addressed while others remained; see below. All things considered, I have increased my score.
> > >
> > > - *many of your criticisms are unfair as they suggest that we should resolve all theoretical issues regarding SGMs*
> > >
> > > I'm not sure where this sentiment comes from. I will try to explain the rationale of my review again, which was aimed at polishing the already solid paper.
> > >
> > > - *In our paper, we have chosen to study the simplest model of generative modeling via reverse SDEs*
> > >
> > > I had issues with the statement "We solved the SGM under assumptions on the score estimates for DDPMs" for the simple reason that what the authors called DDPMs are not the same as in the cited literature. So either the authors change the presentation, which would then slightly weaken the claim, or the authors actually prove the results for DDPMs, which was what I suggested. I think this is quite far from ``resolving all theoretical issues''.
> > >
> > > - *Our goal is not to provide a comprehensive analysis of all SGMs available in the literature but rather to convey our core takeaway and to lay the foundation for future theoretical studies of SGMs, and hence we believe that your suggestions, while relevant, are ultimately out of scope.*
> > >
> > > I wish to point out the limitation of this work so that, perhaps, the authors can either address them or make the presentation clear.
> > >
> > > In any case, I still recommend the authors make it clear that this paper is not about the DDPMs in (Song et al. 2021); I read the revision and I still don't think the message is as clear as it could be.
> > >
> > > - Regarding *One thing that is unclear to me is the scaling issue as follows.*
> > >
> > > I feel like the authors are missing my main point (although I admit that the authors were right about the first term; I apologize for the mistake). I'll try to restate it: Since the performance metrics (in the previous version) are either TV or KL, both scale-invariant, the bounds should ideally also be scale-invariant, which does not seem to be a feature of bounds in the manuscript. I recommend the authors include this as a limitation/future work.
> > >
> > > - *the reference that you sent indeed studies the underdamped Langevin diffusion using the KL divergence... Could you please elaborate on why KL is not the right metric here?*
> > >
> > > Here's an excerpt from the monograph by Villani (slightly rephrased for context; parenthesis mine):
> > >
> > > "In Part III I shall consider fully nonlinear equations, in a scale of Sobolev-type spaces, in presence of a “good” Lyapunov functional... In particular, I shall simplify the proof of the main results in (Desvillettes and Villani 2000).
> > >
> > > Though these three settings are quite different, and far from being unified, there is a unity in the methods that will be used: **construct
> > > a Lyapunov functional by adding carefully chosen lower-order terms to the “natural” Lyapunov functional (i.e., KL divergence)**... "
> > >
> > > - *comparison to De Bortoli*
> > >
> > > This issue is nicely resolved in the revision.
> > >
> > > Thank you for your reading. I'll be happy to engage in any further discussion.

---

> > > > ### Author Response · Authors · 2022-11-29
> > > > **Thank you**
> > > >
> > > > Dear Reviewer,
> > > >
> > > > Thank you for replying and raising your score. We now understand better your points. We will try to address them by adding the following:
> > > >
> > > > - We will state in the abstract that we study the simplest DDPM
> > > >
> > > > - We will add a sentence after the main theorem about the scale non-invariance of our bounds.
> > > >
> > > > - We already conceded that our result on CLD might not say that CLD is not better than Ornstein Uhlenbeck, because CLD might have a statistical advantage (see last paragraph before Section 4). We will add a sentence to also concede that, perhaps, CLD has an advantage that can be seen in other metrics/divergences which are not KL.

---

### Official Review · Reviewer_gx2o · 2022-10-24

**Confidence:** 3
**Correctness:** 4
**Technical Novelty And Significance:** 4
**Empirical Novelty And Significance:** 4
**Recommendation:** 8

**Clarity, Quality, Novelty And Reproducibility:**

This paper is exceptionally clear and well-written, considering that it is a theoretical analysis. I am not an expert in this field, so I am not the best judge of the novelty, but the author's take care to clearly explain the relationship to prior work.

**Strength And Weaknesses:**

Strengths:
- Well-written and presented clearly.
- Clearly discusses the relationship to other work in this area. While I am not an expert in the area, I thought this was especially well-done.
- Bound is in terms of the L2 error of the score estimate.
- Bound does not assume log-concave data distribution.
- Assumptions and limitations of the results are described clearly.
- Explores the consequences of this result with respect to critically damped Langevin diffusion, a variant of the simpler diffusion process.

**Summary Of The Paper:**

This work provides a convergence guarantee for using a score-based diffusion model to sample from an arbitrary distribution. The method has significantly looser assumptions than previous work, and accounts for three sources of error: (1) L2 score estimation error, (2) discretization of the reverse SDE sampling algorithm, and (3) initializing the algorithm from noise rather than the true resulting distribution from the forward diffusion process.



**Summary Of The Review:**

While this is a very technical paper, there is immense interest in diffusion models. I expect this will be of high interest to the community.

---

> ### Author Response · Authors · 2022-11-15
> **Thanks for the positive assessment**
>
> Thank you very much for your kind review. We are glad that you enjoyed our paper.

---

### Official Review · Reviewer_ner6 · 2022-10-25

**Confidence:** 3
**Clarity, Quality, Novelty And Reproducibility:** Clearly written and novel results.
**Correctness:** 4
**Technical Novelty And Significance:** 4
**Empirical Novelty And Significance:** 4
**Recommendation:** 8

**Strength And Weaknesses:**

Strength: The results are pretty impressive given the state of the field.

Weakness: The results shown under the manifold hypothesis seem incomplete.

**Summary Of The Paper:**

This paper provides convergence bounds for score-based models under the assumption that score estimate is $L^2$ accurate. Provided that, the paper derives some remarkable bounds for the score-based sampling methods under pretty weak assumptions.

**Summary Of The Review:**

The paper reads well and presents important results which provides clarity to the field. I have the following comments for the authors:

1) It looks like their rate matches the convergence of Langevin diffusions to the target measure under the LSI assumption - but without any stringent assumption on the target (just good score estimates). However, intuitively (or fundamentally), it was left unclear what actually enables this. Perhaps this is not so surprising, since the forward diffusion starts from actual samples from the target (rather than, like in a classical setting, from an arbitrary point in space) - and provided that the gradient is well estimated, sampling does not require any assumptions. Can authors comment clearly about the differences between their setup and a regular Langevin sampling setup where the initialisations are from arbitrary distributions? This can be perhaps done by assuming no gradient error (exact gradients) and discussing the difference between a Langevin diffusion sampler and a forward-backward score based sampler.

2) It would be also nice if the authors clarified the Remark after Corollary 4. As showed in multiple prior works, the real world data structure supports the manifold hypothesis -- therefore, the impressive convergence results presented in the first part may not apply, as authors pointed out (but still valuable). The remark states that a unified error bound can be obtained in the bounded Lipschitz metric but this was not completed. It would be nice if this is done.

3) Assumption 2 might be OK, assuming finite datasets, however, in a realistic setting where the data stream is observed, this may not hold. Can authors comment if this assumption can be relaxed and at what cost?

---

> ### Author Response · Authors · 2022-11-15
> **Thank you for the review**
>
> Thank you for your comments. We respond to them below.
>
>
> *It looks like their rate matches the convergence of Langevin diffusions to the target measure under the LSI assumption*
>
>
> This is a good question. The reason why SGMs (with zero score estimation error) succeed where Langevin cannot is because **they are fundamentally different**. The forward process is a form of annealing, which has a long history of being applied in non-convex settings; in particular, annealing lowers the energy barrier which causes metastability and slow mixing for the Langevin diffusion. In the spirit, DDPM is related to the proximal sampler, which is the sampling analogue of the proximal operator in optimization. Note that in optimization, applying the proximal operator with a sufficiently large step size can globally minimize even highly non-convex functions. Hence, SGMs can be understood as the sampling version of this fact, where the implementation of the proximal operator is analogous to the implementation of the reversed SDE, and *is achieved through our black-box assumption of accurate score estimation* (which relies, in practice, on the fact that we already have samples from the target, as you wrote).
>
>
> *It would be also nice if the authors clarified the Remark after Corollary 4*
>
>
> Thank you for pointing out this omission. **Yes, we can obtain a unified error bound in the bounded Lipschitz metric**. It is quite immediate from the definition of the BL metric and from the Corollary 4, and we will explicitly state this result in our revision: we obtain $d_{BL}(p_{T-t}, q) \leq \varepsilon$ if the number of iterations is $N = \widetilde\Theta(d^3 R^4 \, (R \vee \sqrt d)^4/\varepsilon^{10})$ and $\varepsilon_{\rm score} \le \widetilde O(\varepsilon)$ (and where we recall that $p_{T-t}$ is the output of the algorithm).
>
>
> *Assumption 2 (moment assumption)*
>
>
> The existence of (a little more than) two moments is an exceedingly weak condition compared to prior works in the literature. First, Assumption 2 (on the moments of the data distribution) is satisfied by any compactly supported data distribution (for example, **Assumption 2 is automatically satisfied in our Corollary 4 on compactly supported data**). Moreover, all distributions arising in practice do indeed have a compact support (or effectively so), since we live in a bounded observable universe. While considering heavy-tailed distributions is an interesting mathematical problem, it is out of scope for the present submission.
>
>
>
> *Thank you for your positive rating of our paper. If you are satisfied with our answers, we would appreciate it if you could raise your score.*

---

> > ### Author Response · Authors · 2022-12-04
> > **Any comment?**
> >
> > Dear Reviewer,
> >
> > Any comment on our rebuttal and the revision of the paper?
> > Did we answer your concerns?
> >
> > Best,
> >
> > Authors

---

### Official Review · Reviewer_uWLa · 2022-10-27

**Confidence:** 3
**Correctness:** 3
**Technical Novelty And Significance:** 3
**Empirical Novelty And Significance:** Not applicable
**Recommendation:** 8

**Clarity, Quality, Novelty And Reproducibility:**

The paper is very clear, despite the complex subject. It goes at great lengths to highlight strengths of the contributions without downplaying the limitations. The appendix is very thorough.

Quality and reproducibility are both acceptable, but less obvious. In particular the quality (which usually for theoretical works corresponds to impact of the result and technical complexity of the novel tools) is made more confusing by the fact that the proof heravily rellies on Chen et al. (2022c) and the bad set approach from (Lee et al., 2022a). While the introduction makes an excellent work in describing how this paper improves over existing results in terms of generality, Sec. 4 could be streamlined to highlight better which novel technical contribution is introduced to improve over existing results. It seems to me that the core of the proof relies on a novel reduction from L_2 to L_infty which then allows to invoke results from (Lee et al., 2022a), but the authors do not clearly state if this reduction already existed in the literature.

Regarding reproducibility, the paper includes a 15 page appendix that heavily relies on two extremely recent results in Chen et al. (2022c) and (Lee et al., 2022a). Chen et al. (2022c) has been peer-reviewed, but for (Lee et al., 2022a) I could not find a peer-reviewed evaluation at all, requiring me to take their results at face value or have to review another 42 page work. This is not an immediate issue with the reproducibility of the result, but it does make it slightly less suitable to a venue with a short publication cycle like ICLR.

**Strength And Weaknesses:**

Despite the strong claims, the paper seems to deliver on most aspects so everything listed in the summary of contribution should be considered as a core strength of the paper.

However each of the strong point presents some slight limitations that could be adressed more clearly.

1) Removing the LSI assumption is a strong result, but ends up introducing new assumptions of smoothness, bounded moment, and of accurate estimation of the score function
1.a) As the authors explain in the paper, this smoothness assumption is not satisfied when the input distribution lies on a manifold, which is probably the same regime where the accurate score function estimation is feasible at all. However most current SGM models also implicitely assume that q has full support, highlighting that this might be a limitation of the whole field. To make the analysis more relevant the failure modes of the theoretical analysis should shed some light on which modifications should be introduced in the models and training procedures to relax this assumption and be able to automatically detect the manifold and avoid this pitfall, but this would be an important contribution by itself. The authors instead give a good second choice by looking at a simplified but realistic model where the distribution lies in a ball around the origin. However the number of iterations grow quite quickly with the radius (e.g. ~R^8) making it vacuous.
1.b) The accurate estimation of the score function is the central point of the paper, but the authors spend very little time explaining how this quantity is computed. Beyond a quick description of score matching, they mostly refer the reader to Vincent 2011, but this source actually highlights how minimizing (13) is actually very hard even after the score matching rewriting, and the authors even mention hardness results at the end of page 2. It would be good to point out some more context on when this estimation problem can be solved in an efficient manner.
1.c) Unlike the other assumptions that are discussed mutliple times in the paper, the moment assumption receives very little attention. For example, the authors should try to justify why their proposed bound of m_2 < d in the discussion of Thm. 2 should hold, especially considering a bound e.g. m_2 < d^2 would result in different value for N and eps_score than the ones reported in the introduction.

**Summary Of The Paper:**

The paper proposes a theoretical analysis of SGMs that extends Chen et al. (2022c) and the bad set approach from (Lee et al., 2022a).

Compared to Chen et al. the analysis is much more general in a number of ways, and compared to Lee et al. the analysis relaxes an LSI assumption and is applicable to a larger class of generative models (CLD).

More concretely, the claimed contributions are:
- more general analysis without LSI assumption and applicable to CLDs
- particularly interesting special case for distributions with a bounded support (which remove the necessity for a restrictive assumption of Lipschitz score estimate)
- tight iteration complexity

**Summary Of The Review:**

Overall I think the paper introduces very strong, novel results that give a good foundation for the study of SGMs. Some limitations are present but they are to be expected due to the short format of the conference. Overall I think it's clearly above the bar for ICLR, but due to the close ties to several recent works it would be good to highlight more what separates the tools used in this paper with previous results.

---

> ### Author Response · Authors · 2022-11-15
> **Thanks for the review**
>
> Thank you for your detailed review. We agree that we can describe some slight limitations more clearly. We will endeavor to do so in our revision. However, we respond to some of your points below.
>
>
>
> *Removing the LSI assumption is a strong result, but ends up introducing new assumptions of smoothness, bounded moment, and of accurate estimation of the score function*
>
> Actually, it is not the case that removing the LSI assumption causes us to add new assumptions. Note that a **bounded second moment is far weaker than an LSI** (an LSI implies the existence of $c > 0$ such that $\exp(c ||.||^2)$ is integrable).
> The smoothness assumption we make is almost the same as the one in Lee et al. 2022, and in fact our assumption is weaker (we do not require the estimated score to be Lipschitz). Besides, the smoothness assumption is not required in our Corollary 4 on compactly supported data.
> Finally, all prior works require accurate estimation of the score functions and our L^2 accurate assumption is one of the weakest assumptions in the literature.
>
>
> - 1a) As you point out, we incur a large polynomial dependency in our result for the compactly supported data case (Corollary 4), though it is not true that our bounds are vacuous as stated in the review. We did not attempt to optimize this bound, as our goal was simply to illustrate that this result (Corollary 4) can be obtained from our main theorem. However, we note that our work is **the first to obtain any polynomial guarantees at all for this setting** (i.e., compactly supported distribution which may not even admit a density, and can be supported on a submanifold), which constitutes a large advance over prior works.
>
>
> - 1b) We reviewed some material on the score estimation problem but we think that the score estimation is a problem itself out of the scope of this paper, and we prefer to leave this problem for future work. Instead, our work addresses the important issue that even with accurate score estimation, prior works did not provide polynomial-time sampling guarantees for SGMs. Finally, let us clarify some confusion about the hardness result: our paper says “sampling is as easy as learning the score” (since we show that learning the score implies sampling) therefore the hardness result is just the contrapositive: “when sampling is hard, learning the score is hard”.
>
>
> - 1c) The existence of (a little more than) two moments is an exceedingly weak condition compared to prior works in the literature. Moreover, all distributions arising in practice do indeed have bounded support (or effectively so) and therefore have two moments. For example, our **moment assumption is automatically satisfied in the Corollary 4 about compactly supported data**. .
>
>
> For many distributions of interest, e.g., the standard Gaussian distribution or product of compactly supported distributions, in fact the second moment $ m_2$ is of order $\sqrt d$. Also, for applications to images in which each coordinate (i.e., pixel) lies in a bounded range $[-1, 1]$, $m_2$ is at most $\sqrt d$. We will add this discussion to our revision.
>
>
>
> *Quality and Reproducibility*
>
>
> Since our original submission, we have discovered that our quantitative bounds can be sharpened and the proof substantially streamlined, see the updated version. In particular, the Girsanov analysis that was the main technical innovation in our original submission turns out to be sufficient on its own, and our proof technique no longer uses the $L^2 \rightarrow L^\inf$ reduction of Lee et al. 2022. This should alleviate your concerns both with regards to quality and to reproducibility.
>
> Besides, note that we do not use any result of Chen et al. 2022c (even if DDPM and the proximal sampler of Chen et al. 2022c are related in the spirit, as you wrote)
>
>
> *We thank you for the nice summary, and we would be grateful if you could raise your score if you are satisfied with our answers.*

---

> > ### Author Response · Authors · 2022-12-04
> > **Any comment?**
> >
> > Dear Reviewer,
> >
> > Any comment on our rebuttal and the revision of the paper?
> > Did we answer your concerns?
> >
> > Best,
> >
> > Authors

---

### Author Response · Authors · 2022-11-15
**To all Reviewers: Revised version with simpler proof and improved quantitative dependencies will be uploaded by Nov 18th**

Thank you to all of the reviewers for your helpful comments. We would like to draw your attention to the fact that since our original submission, we have discovered a simpler proof of our main theorem which also improves the quantitative dependencies of the result; in particular, rather than requiring $\varepsilon_{\rm score}$ (score estimation error) to be smaller than a certain polynomial in $\varepsilon$ (accuracy in TV distance), $L$ (smoothness), and $d$ (dimension), our new result simply requires $\varepsilon_{\rm score}$ to be smaller than $\varepsilon$ up to logarithmic factors. We will update our submission accordingly by Nov 18.

---

### Author Response · Authors · 2022-11-19
**Revision uploaded**

Dear Reviewers and AC,

We uploaded a revised version of our paper answering the reviewers' questions.

Some highlights:

- For **Reviewer uWLa**. New streamlined proof that does not use the  $L^2 \to L^\infty$ reduction argument of Lee et al. 2022 + explanation of why the second moment is of order $\sqrt{d}$.
- For **Reviewer ner6**. New unifying Corollaries for compactly supported data in BL metric (Corollary 5) **and in Wasserstein metric** (Corollary 6)
- For **Reviewer 71Gb**. New Corollary for compactly supported data in Wasserstein metric (Corollary 6) for fair comparison to (De Bortoli 2022)


*We haven't received replies to our answers, but we really hope that the reviewers will check out the revised version.*

---

### Decision · Program_Chairs · 2023-01-20

**Decision:**

Accept: notable-top-5%

**Justification For Why Not Higher Score:**

N/A

**Justification For Why Not Lower Score:**

The paper makes significant contributions to understanding diffusion models. All the reviewers have uniformly rated the paper as a good paper.

**Metareview: Summary, Strengths And Weaknesses:**

This paper derives theoretical convergence guarantees showing that diffusion models can sample from (almost) any data distribution with minimal assumptions on the underlying data distribution, and assuming that an L2-accurate score estimate is learned. Compared to prior works in this space, this submission removes restrictive assumptions that often don't apply to many real-world data distributions and it matches the complexity guarantees for the Langevin diffusion. The paper is extremely well written, making it more accessible to those less familiar with the theoretical analysis of diffusion models. The reviewers have uniformly recognized the significance of this work, and the AC is happy to recommend accept.

**Note From Pc:**

if the above contains the word "oral" or "spotlight" please see: "oral" presentation means -> notable-top-5% and "spotlight" means -> notable-top-25%. As stated in our emails, we are disassociating presentation type from AC recommendations

**Summary Of Ac-Reviewer Meeting:**

N/A